# QuEST: Stable Training of LLMs with 1-Bit Weights and Activations

**Andrei Panferov** [1] **Jiale Chen** [1] **Soroush Tabesh** [1] **Mahdi Nikdan** [1] **Dan Alistarh** [1,2]

## Abstract

One approach to reducing the massive costs of large language models (LLMs) is the use of quantized or sparse representations for training or deployment. While post-training compression methods are very popular, the question of obtaining even more accurate compressed models by *directly training* over such representations, i.e., *Quantization-Aware Training (QAT)*, is still open: for example, a recent study (Kumar et al., 2024) put the "optimal" bit-width at which models can be trained using QAT, while staying accuracy-competitive with standard FP16/BF16 precision, at 8-bits weights and activations. We advance this state-of-the-art via a new method called QuEST, for which we demonstrate optimality at 4-bits and stable convergence as low as 1-bit weights and activations. QuEST achieves this by improving two key aspects of QAT methods: (1) accurate and fast quantization of the (continuous) distributions of weights and activations via Hadamard normalization and MSE-optimal fitting; (2) a new trust gradient estimator based on the idea of explicitly minimizing the error between the noisy gradient computed over quantized states and the "true" (but unknown) full-precision gradient. Experiments on Llama-type architectures show that QuEST induces stable scaling laws across the entire range of hardware-supported precisions, and can be extended to sparse representations. We provide GPU kernel support showing that models produced by QuEST can be executed efficiently. Our code is available at https://github.com/IST-DASLab/QuEST.

## 1. Introduction

The massive computational demands of large language models (LLMs), e.g. (Dubey et al., 2024), have made AI ef-

[1]ISTA [2]Red Hat AI. Correspondence to: Dan Alistarh <dan.alistarh@ist.ac.at>.

*Proceedings of the 42nd International Conference on Machine Learning*, Vancouver, Canada. PMLR 267, 2025. Copyright 2025 by the author(s).

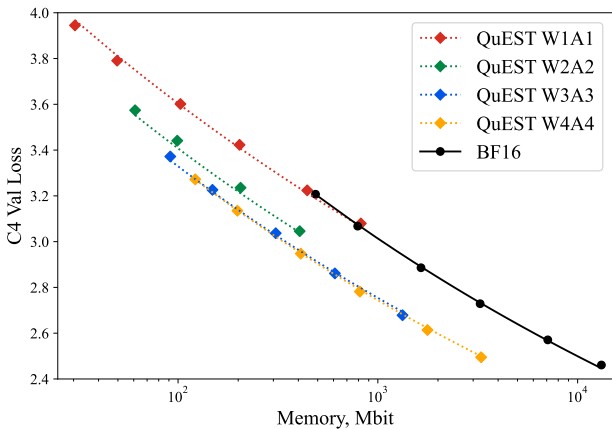

*Figure 1.* The scaling law induced by QuEST when training Llama-family models from 30 to 1.6B parameters on C4, with quantized weights and activations from 1 to 4 bits, in the 100 tokens/parameter regime (harder compression uses proportionally more data at fixed memory). QuEST allows for stable training at 1-bit weights and activations (W1A1), and the QuEST W4A4 model is Pareto-dominant relative to BF16, with lower loss at lower size.

ficiency a critical challenge. One popular pathway to increased efficiency has been *reducing numerical precision*, usually done via *post-training quantization (PTQ)* methods for compressing weights (Frantar et al., 2022; Lin et al., 2024; Chee et al., 2024; Tseng et al., 2024) or both weights and activations (Ashkboos et al., 2023; 2024; Zhao et al., 2023). Quantizing both operands is necessary to leverage hardware support for low-precision multiplications, which extends down to 4-bit (NVIDIA, 2024). However, state-of-the-art PTQ methods are still far from recovering full accuracy for 4-bit precision (Ashkboos et al., 2024; Liu et al., 2024), leaving a gap between computational support and achievable accuracy.

One alternative is *quantization-aware training (QAT)* (Rastegari et al., 2016; Jacob et al., 2018)— where models are trained from scratch with low-precision weights and activations on the forward pass, but with a full-precision backward pass—offering the potential for superior accuracy-vs-compression trade-offs, as gradient optimization can correct compression errors. Despite promising results for weight-only quantization (Wang et al., 2023; Kaushal et al., 2024), it is currently not known whether QAT can produce accurate LLMs with low-bitwidth weights and activations. Here,

the key metric is the *Pareto-optimal frontier*, i.e., the minimal representation size (or inference cost) for the model to achieve a certain accuracy under a fixed data or training budget. Recently, Kumar et al. (2024) identified 8-bit precision as Pareto-optimal for QAT methods on LLMs.

**Contribution.** We present QuEST, a new QAT method that brings the Pareto-optimal frontier to around 4-bit weights and activations and enables stable training at 1-bit precision for both operands. As shown in Figure 1, when data and compute are scaled proportionally to model size, QuEST can train models with 4-bit weights and activations that have superior accuracy relative to BF16 models almost 4x in size.

We achieve this by re-thinking two key aspects of QAT methods: 1) the "forward" step, in which continuous-to-discrete tensor distribution fitting is performed on the forward pass, and 2) the "backward" step, in which gradient estimation is performed over the discrete representation. For the forward step, QuEST works by approximating the "optimal" continuous-to-discrete mapping by first applying a normalizing Hadamard Transform, and then computing an MSE-optimal quantization for the resulting distribution. This replaces the prior "learned" normalization approaches (Choi et al., 2018; Bhalgat et al., 2020).

The key remaining question is how to find an accurate gradient estimator over a weight or activation tensor quantized as above. Here, prior work leverages the Straight-Through Estimator (STE) (Bengio et al., 2013), augmented with learnable components, e.g. (Bhalgat et al., 2020). We propose a different approach called *trust estimation*, which seeks to minimize the difference between the "true" gradient (taken over high-precision weights) and its estimate taken over lower-precision weights and activations. To do this, a trust estimator diminishes the importance of the gradient for some components depending on their *quantization error* on the forward step, following the intuition that entries with large errors lead to significant deviations in the gradient.

Next, we focus on the following question: assuming that training computation is not a limiting factor, what is the "optimal" precision in terms of accuracy-vs-model-size? To address this, we implement QuEST in Pytorch (Paszke et al., 2019) and train Llama-family models (Dubey et al., 2024) of up to 1.6B parameters on up to 160B tokens from the standard C4 dataset (Raffel et al., 2019), across precisions from INT1 to INT8. Results show that QuEST provides stable and accurate convergence across model sizes and precisions down to 1-bit weights and activations. This induces new scaling laws, which we study across model sizes in the large-data (100 tokens/parameter) regime. QuEST leads INT4 weights and activations to be Pareto-optimal in terms of accuracy at a given model size and inference cost, suggesting that the limits of low-precision training are lower than previously thought. In addition, we provide GPU

kernels showing that models produced by QuEST can be run efficiently on commodity hardware.

## 2. Background and Related Work

Hubara et al. (2016) and Rastegari et al. (2016) were among the first to consider training neural networks with highly-compressed internal states, focusing primarily on weight compression. Later work focused on quantization-aware training (QAT) (Jacob et al., 2018; Choi et al., 2018; Esser et al., 2019; Bhalgat et al., 2020) in the form considered here, where the model weights and activations (i.e. the forward pass) are quantized, but the backward pass is performed in full-precision, using variants of the straight-through estimator (STE) (Bengio et al., 2013). (The variant where all states, including gradients, are quantized (Wortsman et al., 2023; Xi et al., 2024) is beyond the scope of this paper.)

Broadly, QAT considers the problem of finding a quantized projection over a *standard-precision tensor* $\mathbf{x}$, representing part of the weights or activations, minimizing output error. For symmetric uniform quantization, the projection onto *the quantized tensor* $\hat{\mathbf{x}}$ is defined as:

$$\hat{\mathbf{x}} = \alpha \cdot \left\lfloor \frac{\text{clip}(\mathbf{x}, \alpha)}{\alpha} \right\rceil, \tag{1}$$

where the clip function performs a clamping operation over the value distribution for all values above the clipping parameter $\alpha$, which also acts as a scaling factor, normalizing values to $\mathbf{x}$ to $[-1, 1]$, and the function $\lfloor \cdot \rceil$ rounds each value to its nearest quantization point, defined as a uniform grid whose granularity depends on the number of available bits $b$ (i.e., $\{-1, \ldots, -\frac{1}{2^b-1}, \frac{1}{2^b-1}, \ldots, 1\}$). Most QAT methods propose to "learn" the factor $\alpha$, for instance, via gradient-based optimization. For example, QAT methods usually keep a standard-precision version $\mathbf{w}$ of the weights; the STE gradient is computed *over the quantized weights* $\widehat{\mathbf{w}}$, and then added to the full-precision accumulator, possibly also updating the clipping factor $\alpha$.

Recent work such as BitNet (Wang et al., 2023; Ma et al., 2024) and Spectra (Kaushal et al., 2024) showed that *weight-only quantization* is viable for small- and medium-scale LLMs. The concurrent work presents BitNet a4.8 (Wang et al., 2024), a hybrid scheme that combines ternary weights with mixed 4- and 8-bit activations, applied selectively to different matrices. In parallel, Kumar et al. (2024) investigated scaling laws for GPT-type models with quantized states, concluding that the "Pareto-optimal" point for current QAT methods is around 8-bit weights and activations.

Prior work by Frantar et al. (2023); Jin et al. (2025) studied scaling laws specifically for sparse foundation models, establishing that the loss can be stably predicted across parameter and data scales when the model weights are sparse.

Recently, Frantar et al. (2025) generalized these laws to unify both sparsity and quantization, allowing to compare the "effective parameter count" for these two types of representations. Our work focuses on improved training methods for highly-compressed representations, leading to improved scaling laws relative to standard dense training, and can be applied to both sparsity and quantization.

# 3. QuEST

**Motivation.** A simple way of describing current QAT methods is that, given a standard-precision tensor $\mathbf{w}$, we first try to get an accurate discrete approximation $\widehat{\mathbf{w}}$ by optimizing parameters such as the clipping factor $\alpha$ in Equation 1 to minimize some loss target, such as the mean-square-error (MSE), and then rely on STE to estimate $\nabla_{\mathbf{w}} L$, the gradient over $\mathbf{w}$, by $\nabla_{\widehat{\mathbf{w}}} L$, the gradient taken w.r.t. the quantized weights $\widehat{\mathbf{w}}$. Yet, the difference between these two gradients, which correlates to the gap in optimization trajectory, could be unbounded, specifically because of large errors in a small subset of entries.

Instead, in this paper, we seek to minimize the "gradient bias," i.e. the difference between the true and discrete gradients, measured, e.g. as

$$\left\| \nabla_{\mathbf{w}} L \ - \ \nabla_{\widehat{\mathbf{w}}} L \right\|_2^2. \tag{2}$$

Prior work on gradient compression (Alistarh et al., 2017; Nadiradze et al., 2021) has identified this quantity as being critical for the convergence of gradient-based optimization algorithms.

Let us define the *quantization error* for each entry $w_k$ as $\text{err}_k = \left| w_k - \hat{w}_k \right|$. We can partition the weight indices $k$ based on whether the quantization error $\text{err}_k$ is smaller or larger than some "trust factor" threshold $T$. Denote:

$$S_{\text{small}} = \{ k : \text{err}_k \le T\}, \quad S_{\text{large}} = \{ k : \text{err}_k > T\}.$$

Then, the squared gradient difference in (2) decomposes as:

$$\underbrace{\sum_{k \in S_{\text{small}}} (\nabla_{\mathbf{w}} L_k \ - \ \nabla_{\widehat{\mathbf{w}}} L_k)^2}_{(\star)} + \underbrace{\sum_{k \in S_{\text{large}}} (\nabla_{\mathbf{w}} L_k \ - \ \nabla_{\widehat{\mathbf{w}}} L_k)^2}_{(\star\star)}.$$

Assuming that the loss $L$ is $\gamma$-smooth, the $(\star)$ "small error" term would be upper bounded by $\gamma^2 T^2 |S_{\text{small}}|$. Intuitively, this term is minimized in a standard QAT method's "distribution fitting" step. Yet, distribution fitting does not address the "large error" term $(\star\star)$: specifically, outlier entries clipped in the fitting step can lead to extremely large gradient estimation errors.

QuEST takes this into account by balancing estimation errors due to minor but persistent quantization errors in $(\star)$, with the significant "outlier" errors incorporated by term $(\star\star)$. For this, we propose an efficient fitting mechanism that minimizes persistent errors, coupled with a "trust" gradient estimator step aimed at bounding outlier errors.

## 3.1. Step 1: Distribution Fitting

While optimizing the quantization grid to best fit the underlying tensor is a core idea across all quantization methods, PTQ methods traditionally use more complex and computationally heavy approaches (Dettmers et al., 2024; Malinovskii et al., 2024). In contrast, QAT methods rely on backpropagation through the scaling factor for error-correction (Esser et al., 2019; Bhalgat et al., 2020) while performing re-fitting. To avoid backpropagation errors impacting the forward pass, we do not use backpropagation for distribution fitting. Instead, we start from the empirical observation that the distribution of weights and activations during LLM training is sub-Gaussian but with long tails (Dettmers et al., 2022; 2023).

**Gaussian Fitting.** Specifically, we choose to optimize the grid to explicitly fit a Gaussian distribution with the same parametrization as the empirical distribution of the underlying tensor $\mathbf{x}$. Concretely, we use *root mean square (RMS)* normalization to first align the empirical distribution of $\mathbf{x}$ with a $\mathcal{N}(0, 1)$ Gaussian distribution (Frantar et al., 2025). We then perform the projection operation with the scale $\alpha^*$ chosen to minimize the $L_2$ error resulting from projecting $\mathcal{N}(0, 1)$. Formally:

$$\widehat{\mathbf{x}} = \alpha^* \cdot \text{RMS}(\mathbf{x}) \cdot \left\lfloor \frac{\text{clip}\left(\mathbf{x}/\text{RMS}(\mathbf{x}), \alpha^*\right)}{\alpha^*} \right\rceil =$$
$$:= \text{proj}_{\alpha^*}(\mathbf{x}), \text{ where}$$
$$\alpha^* := \underset{\alpha \in \mathbb{R}}{\arg \min} \, \mathbb{E}_{\xi \sim \mathcal{N}(0,1)} \left\| \xi - \alpha \cdot \left\lfloor \frac{\text{clip}(\xi, \alpha)}{\alpha} \right\rceil \right\|_2^2$$

is the MSE-optimal scaling factor. If $\mathbf{x}$ were Gaussian-distributed, this would produce an MSE-optimal projection.

**Hadamard Preprocessing.** Yet, the natural distribution of tensor values may not be Gaussian, especially given the emergence of outlier values (Dettmers et al., 2022; Nrusimha et al., 2024). To mitigate this, we add a Hadamard Transform (HT) step *before* Gaussian Fitting. Thus, our forward pass projection becomes:

$$\hat{\mathbf{x}}_h = \text{proj}_{\alpha^*} \text{HT}(\mathbf{x}). \tag{3}$$

In other words, we transform the target tensor via multiplication with a Hadamard matrix of appropriate shape, applied along the matrix-multiplication dimension, and then project it to an MSE-optimal grid in the Hadamard domain.

Here, we leverage 1) the fact that, roughly, multiplication of a matrix with the Hadamard Transform leads the weight distribution to better match a Gaussian (Ailon & Chazelle, 2009; Suresh et al., 2017); 2) the existence of fast Hadamard multiplication kernels (Tri Dao), and 3) the fact that the HT is orthogonal, so it can be easily inverted. While this HT effect has been utilized in PTQ (Tseng et al., 2024; Ashkboos et al., 2024; Malinovskii et al., 2024) and distributed optimization (Vargaftik et al., 2021; 2022), we believe we are the first to harness it for QAT.

## 3.2. Step 2: Trust Gradient Estimation

Next, we focus on the backward pass. For simplicity, we first describe the variant without the Hadamard Transform step and then integrate this component.

**Trust Estimators for the Basic Projection.** First, assume that $\hat{\mathbf{x}} = \mathrm{proj}_{\alpha^*}(\mathbf{x})$. Since the projection operation $\lfloor x \rceil$, is not differentiable w.r.t. $x$, we need a robust way to estimate our gradient. Expressed as an operator, STE can be written as $\frac{\partial}{\partial \mathbf{x}} \approx \frac{\partial}{\partial \lfloor \mathbf{x} \rceil}$ during the backward pass, allowing gradients to propagate through the network, but can lead to large errors due to components with large quantization error.

Specifically, the factor $\alpha^*$, chosen to minimize the weight fitting error, acts as a natural scale for how far off their real value the majority of quantized values can be: for values below the scaling factor, this error is not larger than $T = \frac{\alpha^*}{2^b - 1}$, the half-width of a quantization interval. This gives a natural bound for the $(\star)$ term in our analysis of Equation 2.

To bound the second term $(\star\star)$, we choose to *not trust* the gradient estimations for weights with large errors $\{\nabla_{\hat{\mathbf{w}}} L_k : k \in S_{\mathrm{large}}\}$. Choosing $T = \frac{\alpha^*}{2^b - 1}$ and masking gradients for elements in $S_{\mathrm{large}}$ we obtain the gradient operator:

$$\frac{\partial}{\partial \mathbf{x}} \approx \mathbf{I}_{|\hat{\mathbf{x}} - \mathbf{x}| \leq T} \odot \frac{\partial}{\partial \hat{\mathbf{x}}} := M_{\alpha^*}(\mathbf{x}; \hat{\mathbf{x}}) \odot \frac{\partial}{\partial \hat{\mathbf{x}}},$$

where $\mathbf{I}_{|\hat{\mathbf{x}} - \mathbf{x}| \leq T}$ is the standard indicator operator. We will refer to $M_{\alpha^*}$ as the "trust mask"; this gradient estimation operator will be called the **trust estimator**.

**Trust Estimators for the Hadamard Projection.** We now interface the trust estimator with the Hadamard Transform (HT) and its inverse (IHT) to obtain the following forward scheme: $\mathbf{x}_h = \mathrm{HT}(\mathbf{x})$ and $\hat{\mathbf{x}}_h = \mathrm{proj}_{\alpha^*} \mathbf{x}_h$. Then, the natural approach is to perform trust estimation directly in the Hadamard domain, where quantization takes place:

$$\frac{\partial}{\partial \mathbf{x}} \approx \mathrm{IHT}\left(M_{\alpha^*}(\mathbf{x}_h; \hat{\mathbf{x}}_h) \odot \frac{\partial}{\partial \hat{\mathbf{x}}_h}\right).$$

In other words, after deriving the trust mask w.r.t. distribution fitting in the Hadamard domain, we apply the result-

---

**Algorithm 1** QuEST Training Forward

1: **Input:** Input activations $\mathbf{x}$, row-major weight $\mathbf{w}$
2: $\mathbf{x}_h = \mathrm{HT}(\mathbf{x})$
3: $\hat{\mathbf{x}}_h = \mathrm{proj}_{\alpha^*} \mathbf{x}_h$
4: $\mathbf{w}_h = \mathrm{HT}(\mathbf{w})$
5: $\hat{\mathbf{w}}_h = \mathrm{proj}_{\alpha^*} \mathbf{w}_h$
6: $\mathbf{y} = \hat{\mathbf{x}}_h \hat{\mathbf{w}}_h^T$
7: **Return:** $\mathbf{y}, \hat{\mathbf{x}}_h, \hat{\mathbf{w}}_h, M_{\alpha^*}(\mathbf{x}_h; \hat{\mathbf{x}}_h), M_{\alpha^*}(\mathbf{w}_h; \hat{\mathbf{w}}_h)$

---

ing mask $M_{\alpha^*}(\mathbf{x}_h; \hat{\mathbf{x}}_h)$ onto the gradient w.r.t. quantized weights *in the Hadamard domain*.

**Gradient Effects.** Notice that, in the absence of the HT or regularization effects (e.g., weight decay), the "untrusted" weights in $S_{\mathrm{large}}$ would receive no gradient and may be permanently removed from optimization. Yet, the addition of the HT means that the trust mask is *no longer binary* in the "standard" domain, allowing for gradient flow towards *all model weights*. We validated this effect empirically by observing that the HT reduced the final cardinality of the "untrusted" weights set $S_{\mathrm{large}}$ by $\approx 4$x, aligning it with the number of values we would expect to be outside the "trust set" at every step, for weights from a normal distribution. This is investigated in more depth in Appendix A.1.

## 3.3. Discussion

**Implementation.** In practice, we use identical Hadamard Transforms along the matrix-multiplication dimension for both the weights $\mathbf{w}$ and the activations $\mathbf{x}$. Since the Hadamard Transform is unitary, the quantized matrix multiplication output $\mathbf{y} = \hat{\mathbf{x}} \hat{\mathbf{w}}^T$ is aligned with the full precision output $\mathbf{x} \mathbf{w}^T$ it approximates. The algorithm 1 describes the forward pass over a linear layer actively quantized with QuEST for a row-major weight representation.

The algorithm 2 describes the backward pass over the same layer using the quantized weight and activations from the forward pass as well as error gradient w.r.t $\mathbf{y}$. We note that, although the backward computation is performed w.r.t. the *quantized* weights and activations, the multiplications and gradient operands are performed in standard 16-bit precision.

---

**Algorithm 2** QuEST Training Backward

1: **Input:** $\frac{\partial L}{\partial \mathbf{y}}, \hat{\mathbf{x}}_h, \hat{\mathbf{w}}_h, M_{\alpha^*}(\mathbf{x}_h; \hat{\mathbf{x}}_h), M_{\alpha^*}(\mathbf{w}_h; \hat{\mathbf{w}}_h)$
2: $\frac{\partial L}{\partial \hat{\mathbf{x}}_h} = \frac{\partial L}{\partial \mathbf{y}} \hat{\mathbf{w}}_h$
3: $\frac{\partial L}{\partial \mathbf{x}} = \mathrm{IHT}\left(M_{\alpha^*}(\mathbf{x}_h; \hat{\mathbf{x}}_h) \odot \frac{\partial L}{\partial \hat{\mathbf{x}}_h}\right)$
4: $\frac{\partial L}{\partial \hat{\mathbf{w}}_h} = \hat{\mathbf{x}}_h^T \frac{\partial L}{\partial \mathbf{y}}$
5: $\frac{\partial L}{\partial \mathbf{w}} = \mathrm{IHT}\left(M_{\alpha^*}(\mathbf{w}_h; \hat{\mathbf{w}}_h) \odot \frac{\partial L}{\partial \hat{\mathbf{w}}_h}\right)$
6: **Return:** $\frac{\partial L}{\partial \mathbf{x}}, \frac{\partial L}{\partial \mathbf{w}}$

---

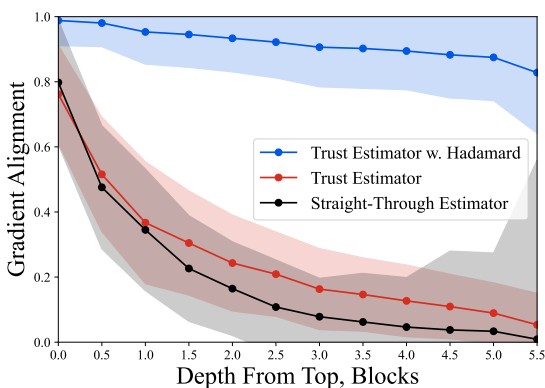

*Figure 2.* Gradient alignment comparison for a 30M Llama model after training on 2.7B tokens in 8-bit precision.

**Training Complexity.** In total, during training, for each original matrix multiplication (e.g., $\mathbf{xw}^T$), we need only two Hadamard Transforms on the forward pass and two Inverse Hadamard transforms on the backward pass.

For a Transformer model (Vaswani, 2017) with $d$ blocks and hidden dimension $h$, and a batch containing $b$ tokens, the MatMul complexity of the forward pass can be estimated as: $b \times d \times h^2$. Then, the asymptotic cost of the Hadamard Transform is the quantity $b \times d \times h \times \log h + d \times h^2 \times \log h$, which is asymptotically negligible with $b > \log h$.

**Activation Effects.** It is well-known (Choi et al., 2018) that activation quantization has major impact on training, possibly due to compounding with model depth. To test the effect of different gradient estimators on backpropagation, we empirically examine "gradient quality" as follows: we calculate intermediate gradients $\nabla_{\hat{\mathbf{a}}^\ell} L$ with respect to activations after the $\ell$-th Transformer block. For the same input, we disable activations quantization and calculate the "true" gradients $\nabla_{\mathbf{a}^\ell} L$. We then define the "gradient alignment" as the cosine similarity between gradients: $\Xi(\nabla_{\hat{\mathbf{a}}^\ell} L, \nabla_{\mathbf{a}^\ell} L) = (\nabla_{\hat{\mathbf{a}}^\ell} L \cdot \nabla_{\mathbf{a}^\ell} L)/(\|\nabla_{\hat{\mathbf{a}}^\ell} L\|_2 \|\nabla_{\mathbf{a}^\ell} L\|_2)$.

While low similarity does not necessarily indicate poor gradient estimation (as the quantized forward pass might have utilized slightly different pathways, leading to discrepancy), high similarity clearly indicates that the estimator produces "high-quality" gradients relative to full precision. Figure 2 compares the gradient alignment for the STE relative to QuEST, with and without the HT. QuEST leads to remarkably-high and well-concentrated alignment ($\geq 0.8$), even at larger depths. By contrast, standard trust estimation degrades alignment with depth but has good concentration, whereas the STE has *poor alignment and high variance*.

**The 1-bit Case.** In our original trust estimation formulation, we proposed to set the trust factor as half the quantization interval, $T = \frac{\alpha^*}{2^b-1}$. Thus, the trust regions increase exponentially as the bitwidth decreases. In particular, for 1-bit

weights and activations, QuEST will suffer from trust regions that extend out of the grid by a whole $\alpha^\star$. To fix this, we reduce the size of the "outermost" trust regions, outside the clipping factor, by a scaling factor $s$. Through small-scale experiments, we determined the optimal value of $s$ to be $s^\star \approx 1.30$. We use this scaling factor for all the 1-bit QuEST runs in this paper (unless stated otherwise). This modification is necessary (and leads to an improvement) only in the extreme 1-bit compression regime. This is discussed further in Appendix A.2.

## 4. Experimental Validation

### 4.1. Implementation Details

**Models and Hyperparameters.** We tested our method on pre-training decoder-only Transformers (Vaswani, 2017) following the Llama architecture (Touvron et al., 2023), in the range of 30, 50, 100, 200, 430 and 800 million non-embedding parameters. Please see Appendix B.1 for architecture and hyper-parameter details. We trained all models on tokens from the C4 (Dodge et al., 2021) dataset, tokenized with the Llama 2 tokenizer. We used the AdamW (Loshchilov & Hutter, 2019) optimizer with a cosine learning rate schedule and a 10% warmup period, with gradient clipping (1.0 threshold, decoupled weight decay of 0.1). We identified the learning rate optimally for a 50M FP16 model via a learning-rate sweep. For other models, as standard, we scale the learning rate inverse-proportionally to the number of non-embedding parameters. We reuse the exact learning rates for all QuEST training runs. Please see https://github.com/IST-DASLab/QuEST for a reference implementation.

Unless stated otherwise, we train every model on a number of tokens equal to 100x its number of "free" parameters, e.g., 10B tokens for a Llama 100M model, regardless of precision. This allows us to explore the data-saturation regime. We aim for comparisons that are iso-size: That is, to match the size / FLOPs of a 100M FP16 Llama model (trained on 10B parameters), we will train a 400M-parameter model with 4-bit weights and activations, using 40B total tokens. This allows us to explore accuracy for fixed model sizes, across compression ratios (see Figure 1). We discuss different $D/N$ regimes in Appendix C.2.

### 4.2. Comparison to Prior QAT Methods

We compare QuEST to: STE; LSQ (Esser et al., 2019), a widely used QAT baseline; a QAT extension of QuaRot (Ashkboos et al., 2024), a method similar to QuEST but with AbsMax scaling instead of proper distribution matching; and AdaBin (Tu et al., 2022), a specialized W1A1 training method. The results, presented in Table 1, indicate that QuEST outperform all existing methods, including spe-

| Model size | Method | W4A4 | W3A3 | W2A2 | W1A1 |
|---|---|---|---|---|---|
| 30M | STE | 3.792 | 4.449 | 4.793 | 5.256 |
| | QuaRot | 3.338 | 3.612 | 4.481 | 4.932 |
| | LSQ | 3.315 | 3.410 | 3.598 | 3.991 |
| | AdaBin | – | – | – | 3.988 |
| | **QuEST** | **3.272** | **3.372** | **3.574** | **3.945** |
| 50M | STE | 4.040 | 4.542 | 5.162 | 6.867 |
| | QuaRot | 3.201 | 3.695 | 4.566 | 5.007 |
| | LSQ | 3.240 | 3.290 | 3.501 | 3.862 |
| | AdaBin | – | – | – | 3.843 |
| | **QuEST** | **3.135** | **3.226** | **3.441** | **3.791** |

*Table 1.* C4 validation loss comparison across bit-widths and model sizes for STE, a QAT extension of QuaRot, LSQ, AdaBin and QuEST. AdaBin is only defined in the binary case.

cialized ones, across all tested bitwidths. We perform a more elaborate numerical comparison in the next section.

### 4.3. Scaling Laws

**Background.** Hoffmann et al. (2022) proposed to model loss scaling as a function of the number of parameters in the model $N$ and the number of tokens $D$ it was trained on, in the form of parametric function:

$$L(N, D) = \frac{A}{N^\alpha} + \frac{B}{D^\beta} + E, \qquad (4)$$

where $A$, $B$, $E$, $\alpha$, and $\beta$ are the scaling law parameters that can be fit empirically. Following Frantar et al. (2025), we modify this formula assuming that the training precision $P$ only affects the parameter count $N$ as a multiplicative factor $\mathrm{eff}(P)$, which, for a given quantization method, depends only on the training precision:

$$L(N, D, P) = \frac{A}{(N \cdot \mathrm{eff}(P))^\alpha} + \frac{B}{D^\beta} + E. \qquad (5)$$

If we take $\mathrm{eff}(16) = 1.0$, we recover the law in Equation 4.

**Fitting process.** To estimate $A$, $B$, $E$, $\alpha$, $\beta$ and $\mathrm{eff}(P)$ for every quantization precision $P$ we need, we fit this parametric function by minimizing the Huber loss (Huber, 1964) between the predicted and the observed log loss. Our process is detailed in the Appendix, and closely follows the setup of Hoffmann et al. (2022), including the grid search and the loss hyper-parameters.

Specifically, we fit the model on the range of parameters $P \in \{1, 2, 3, 4, 16\}$, $N \in \{30, 50, 100, 200, 430, 800\} \times 10^6$ and $D = 100 \times N$. The resulting fit is presented on Figure 1. To capture a larger range of $D$, we fit the model on additional runs with $P \in \{2, 3, 4\}$, $N \in \{30, 50, 100\} \times 10^6$ and $D/N \in \{25, 50\}$. We additionally fit the extensions

| $P$ | 1 | 2 | 3 | 4 | 8 | 16 |
|---|---|---|---|---|---|---|
| QuEST | 0.02 | 0.16 | 0.43 | 0.70 | 1.02 | 1.00 |
| LSQ | 0.02 | 0.12 | 0.32 | 0.56 | 0.87 | 1.00 |

*Table 2.* Fitted scaling-law parameter efficiencies $\mathrm{eff}(P)$.

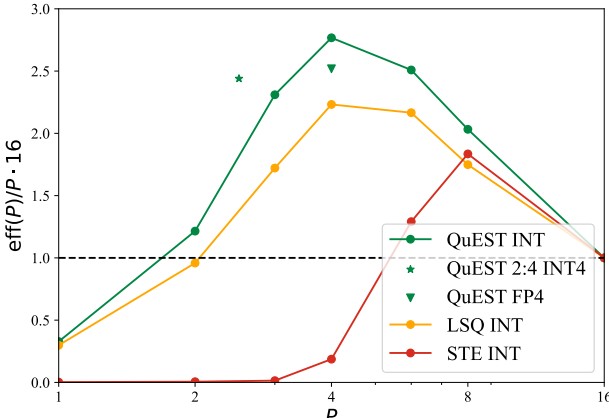

*Figure 3.* Illustration of the efficiency factors $\mathrm{eff}(P)/P$, arising from our analysis, for different numerical precisions $P$, formats (INT, FP, INT+sparse) and methods. Higher is better. QuEST INT4 appears to have the highest efficiency.

of our method described in Sections 5 and 4.6. Appendix Figure 12 illustrates the quality-of-fit.

**Results.** The overall results were presented in Figure 1, illustrating loss vs. model size. First, we observe that, remarkably, QuEST provides stable training down to 1-bit weights and activations, across model sizes, following a stable scaling law. Second, examining the Pareto frontier, we observe that 4-bit precision is slightly superior to 3-bit, and consistently outperforms all higher precisions. Overall, these results show that QuEST can lead to stable scaling laws, which consistently improve upon prior results (Kumar et al., 2024), moving the Pareto-optimal line to around 4-bit.

### 4.4. Finding the "Optimal" Precision

**The Overtraining (OT) regime.** The goal of a standard scaling law (Equation 4) is to determine the "optimal" model size $N$ and training duration $D$ under fixed pre-training compute $C = 6ND$. For instance, Hoffmann et al. (2022) estimated the "Chinchilla-optimal" ratio to be around $D/N \approx 20$. Yet, it is now common to train (often smaller) models way beyond this ratio, effectively spending additional training compute (relative to "optimal") to minimize deployment costs by executing a smaller model. For example, recent models are trained with $D/N \geq 1000$ (Dubey et al., 2024; Team et al., 2024). With test-time compute (Snell et al., 2024), there is an incentive to increase

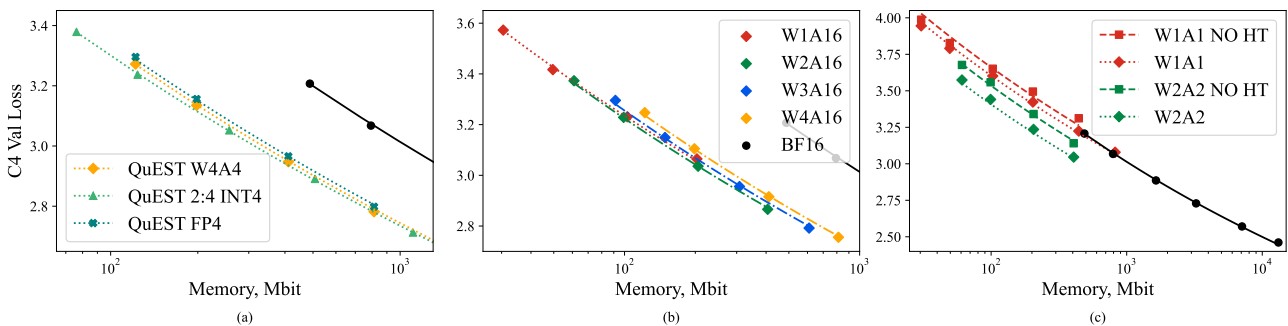

*Figure 4.* Additional scaling laws induced by QuEST: **(a, left)** compares INT, FP, and INT+sparse formats at 4-bit precision, **(b, middle)** shows the scaling laws for weight-only quantization, where 2-bit appears to be Pareto-dominant, while **(c, right)** shows that trust estimation benefits significantly from Hadamard normalization.

this even further. If we extrapolate and take $D/N \to \infty$, Equation 5 takes the simplified form:

$$L_{OT}(N, P) = \frac{A}{(N \cdot \text{eff}(P))^\alpha} + E. \qquad (6)$$

We refer to this as the "overtraining" (OT) regime, where the training compute is less relevant, and is only bounded by factors such as the available amount of filtered training data. The focus is on minimizing *runtime/inference* compute, measured for example by model latency. This problem can be formulated as finding the optimal model size $N$ and precision $P$ that minimizes a certain runtime compute limit.

**Runtime Cost Estimate.** Since we focus on quantizing both weights and activations, the matrix multiplications can be performed directly in lower-precision, providing linear speedups in the precision $P$ (Abdelkhalik et al., 2022). As such, we can roughly estimate the runtime cost, up to constants, as the precision-weighted number of basic operations (FLOPs) in a forward pass $F = NP$. Then, the problem of minimizing loss while staying within a certain runtime (FLOP) constraint can be re-written as:

$$\min_{N,P} L_{OT}(N, P) = \frac{A}{\left(F \cdot \frac{\text{eff}(P)}{P}\right)^\alpha} + E \ \text{ s.t. } F \leq F_{\max}.$$

From this formulation, if we fix $F \leq F_{\max}$, maximizing $\frac{\text{eff}(P)}{P}$ becomes the key factor that influences the "optimal" pre-training precision in the OT regime. Recall that we can estimate $\text{eff}(P)$ from the empirical scaling law (obtained in Section 4.3 and shown in Table 2). Thus, we can calculate $\frac{\text{eff}(P)}{P}$ for any precision. Figure 3 suggests that 4-bit appears to be the optimal pre-training precision in this regime. Additionally fitting $\text{eff}(P)$ for selected baselines and plotting them on the same figure, one can see the dominance

of QuEST across all bitwidths with gaps aroung 50% of baseline efficiency around the optimal precision.

### 4.5. Extensions to Different Formats

**The FP4 Format**. We can use the same framework to compare the "effective parameter count" for INT, INT + sparse, and the lower-precision FP format supported by NVIDIA Blackwell (NVIDIA, 2024). QuEST can be extended to this data type by replacing the $\lfloor \cdot \rceil$ rounding operation with rounding to the FP4 grid $\lfloor \cdot \rceil_{\text{FP4}}$ scaled to fit the same $[-1, 1]$ interval. The optimal scaling factor $\alpha^*_{\text{FP4}}$ would be defined by simply replacing $\lfloor \cdot \rceil$ with $\lfloor \cdot \rceil_{\text{FP4}}$ in the original definition. We choose the trust factor $T$ for $M_{\alpha^*}(\mathbf{x}; \hat{\mathbf{x}}) = \mathbf{I}_{|\hat{\mathbf{x}}-\mathbf{x}| \leq T}$ as the largest half-interval of the FP4 grid.

To determine the $\text{eff}(P)$ parameter for FP4, we train 30, 50, 100, and 200M models with QuEST in FP4 precision and aggregate results in Figure 4(a), comparing them with the original uniform grid results. We observe that FP4 performs slightly worse than INT4. We also fit FP4 with the scaling law in Equation (5) and present the resulting $\text{eff}(P)/P$ in Figure 3 (red dot). The results show that, indeed, FP has lower parameter efficiency than INT at 4-bit precision. We hypothesize that this is correlated with the fact that, when clipping is allowed, FP4 has higher MSE than INT4 when fitting Gaussian-distributed data.

**Extension to sparsity.** QuEST can also be extended to sparsity. Then, the trust estimator will mask out sparsified elements with absolute value above the trust mask; specifically, this covers the majority of sparsified elements, except for the small elements within $\left[-\frac{\alpha^*}{2^b-1}, +\frac{\alpha^*}{2^b-1}\right]$. In practice, we still keep the whole weight matrix in full precision during training. On the forward pass, we first sparsify and then quantize. On the backward pass, we apply the trust mask as usual.

Figure 4(a) illustrates the scaling law induced by the 50%

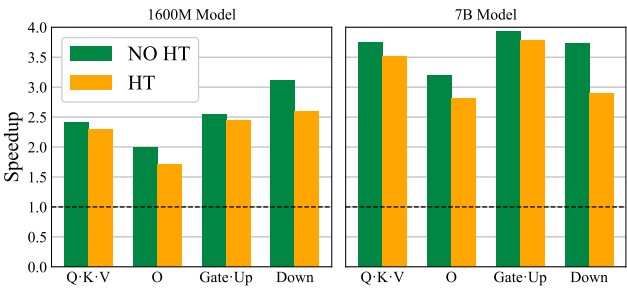

*Figure 5.* Per-layer speedups for QuEST INT4 vs BF16, on a single RTX 4090 GPU. The results take into account quantization/dequantization costs for QuEST, and include the cost of the Hadamard transform (orange bar). We present results for the 1.6B 4-bit QuEST model we trained, as well as inference speedups for a proportional 7B-parameter model.

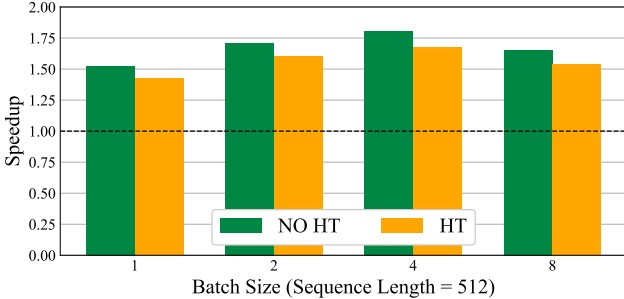

*Figure 6.* End-to-end prefill speedups for QuEST INT4 vs BF16, across different batch sizes, using the 1.6B parameter model on a single RTX 4090 GPU. As expected, QuEST is most effective for larger batch sizes, where the workload is more compute-bound.

sparse + INT4 of NVIDIA Ampere (Abdelkhalik et al., 2022), while Figure 3 (green dot) shows its parameter efficiency relative to INT and FP. With QuEST, this format can provide better scaling than FP4, but slightly inferior to INT4. (While this format is known as 2:4 sparsity, for INT4 + 2:4 it requires a 4:8 mask with some additional constraints.)

### 4.6. Additional Experiments

**Weight-only quantization.** In addition to the comparison with the baseline presented in Section 4.2, we present full scaling for weight-only QuEST quantized training. We train models with 30, 50, 100, and 200 million parameters in 1,2,3, and 4 bits in the same general setup as Figure 1. The results in Figure 4(b) show that our approach leads to stable scaling laws in the weight-only case as well. Interestingly, here 2-bit weights appear to be Pareto-dominant, while 1-bit is surprisingly competitive with 3-bit weights.

**Hadamard ablation.** Finally, we examine the impact of the Hadamard transform by removing it while maintaining the trust technique, as described in Section 3.2. In Figure 4(c), we present the results in the same setup as Figure 1 for a simplified trust scheme without the Hadamard Transform. Specifically, 1) training remains stable across all precisions, although W1A1 is now inferior to BF16; 2) W4A4 remains Pareto-dominant, suggesting that the Hadamard transform improves the coefficients but does not alter the scaling laws.

## 5. GPU Execution Support for QuEST Models

**Kernel Overview.** Finally, we describe GPU kernel support. Our forward-pass pipeline for the quantized linear layer in QuEST consists of three main stages: (1) applying the Hadamard transformation to the BF16 activations, (2) quantizing the BF16 activations into INT4 and packing them into the low-precision format, and (3) performing INT4 ma-

trix multiplication on the quantized activations and weights, followed by dequantization of the result back to BF16.

For the first stage, we utilize an existing Hadamard kernel (Tri Dao). We developed a custom Triton kernel for the second stage to fuse the quantization and data formatting. This kernel computes MSE-optimal group scales and performs centered quantization on the activations. It also packs the INT4 elements into UINT8, with additional intermediate results prepared for matrix multiplication and dequantization. The third stage involves fused matrix multiplication and dequantization using our enhanced CUTLASS kernel. In this stage, both activations and weights are read and processed as integers to exploit the higher GPU throughput. The results are then dequantized back to BF16 within the same kernel. We also apply CUDA Graph end-to-end to further reduce the kernel launching overhead.

To optimize GEMM performance, we carefully tuned the CUDA thread-block and warp tile sizes and leveraged the high levels of the memory hierarchy to fuse the dequantization step before writing the results back to Global Memory in a custom CUTLASS *epilogue*. By performing dequantization at the register level, we minimize data movement, reduce GMEM memory access overhead, and minimize the number of kernel launches.

**Runtime Results.** The per-layer speedups achievable using our kernel at 4-bit precision, relative to 16-bit MatMuls, are illustrated in Figure 5. We provide a breakdown across layers of the same shape, for 1.6B (which we have already trained), and a proportionally-scaled 7B model (which we plan to train in future work). These measurements include all auxiliary overheads (e.g. quantization/dequantization) for QuEST; in addition, we separate out the performance impact of the Hadamard transform.

For the smaller 1.6B model, the per-layer speedups vary between 1.2× (on the smallest layers, with Hadamard) and 2.4× (largest down-projection layer, no Hadamard). The

largest overhead of the Hadamard transform, of around 30%, is on the down-projection layer, which presents the largest dimension for the Hadamard. The speedups increase significantly (2.3-3.9×) when we move to the 7B-parameter model, as the MatMuls are much more expensive. Figure 6 shows the end-to-end inference performance at 1.6B using our kernels vs. the BF16 baseline, showing speedups of 1.3-1.5× in the less memory-bound regime.

## 6. Discussion and Future Work

We introduced QuEST, a new QAT method that achieves stable LLM training of in extremely low precision (down to 1-bit) weights and activations. Our results demonstrate that, if data and compute are appropriately scaled, 4-bit models can outperform standard-precision baselines in terms of accuracy and inference cost, suggesting that the fundamental limits of low-precision QAT are much lower than previously thought. Further, our analysis provides new insights into the relationship between training precision and model efficiency, suggesting that low-precision may be a good target for large-scale training runs in the overtrained regime. Third, we have shown that our approach can lead to inference speedups.

Several promising directions emerge for future work. First, while we demonstrated QuEST's effectiveness up to 1.6B parameters, its scaling behavior for much larger models is an interesting direction we plan to pursue in future work. Second, our work focused primarily on decoder-only architectures; extending QuEST to encoder-decoder models and other architectures could broaden its applicability.

## Acknowledgements

The authors would like to thank Roberto Lopez Castro for key contributions to the GPU kernel implementations presented in Section 5. We would also like to thank Elias Frantar for useful preliminary discussions. Further, the authors would like to thank Martin Jaggi for his generous support during the development of this project. This research was funded in part by the Austrian Science Fund (FWF) 10.55776/COE12.

## Impact Statement

This paper presents work whose goal is to advance the field of Machine Learning. There are many potential societal consequences of our work, none of which we feel must be specifically highlighted here.

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

# A. Additional "Trust" Details

## A.1. Trust Mask Analysis

For the purposes of weight trust masks interpretation, we trained a 30M model over 3B tokens (11,444 iterations at bs=512) with QuEST weights and activations quantization to 8-bit with and without the Hadamard Transform (HT). We logged the trust masks every 500 iterations. Figure 7 shows the fraction of masked weights. We can see that adding the HT leads to an ≈4x decrease in the amount of masked values, corresponding to the fraction of expected clipped weights for a standard normal distribution. We can also see that without the HT the fraction deviates significantly from the expected fraction under the assumption of weights normality.

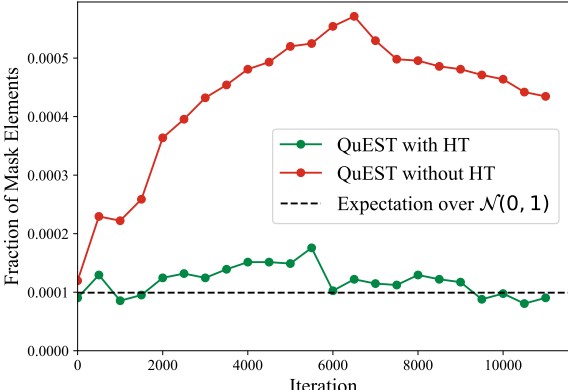

*Figure 7.* Fraction of weights for which $M_{\alpha^*} = 0$ as a function of number of training iterations for a 30M model trained with QuEST.

Moreover, we looked at the percentage of masked elements at a fixed iteration in the past, that remain masked at a fixed later iteration. We plot these percentages in Figure 8. As we can see, for the run without the HT, around 69% of masked elements at iteration 6000 (roughly halfway through training) remain masked at iteration 10000 (towards the end of the training). This percentage is more than twice as small for the run with the HT at 30%. This implies that the HT makes masks less persistent, as expected. In addition, we note that weight decay is applied on all weights (including masked ones). Thus, a masked weight will slowly decay until it may "exit" the masked interval, obtaining gradient again.

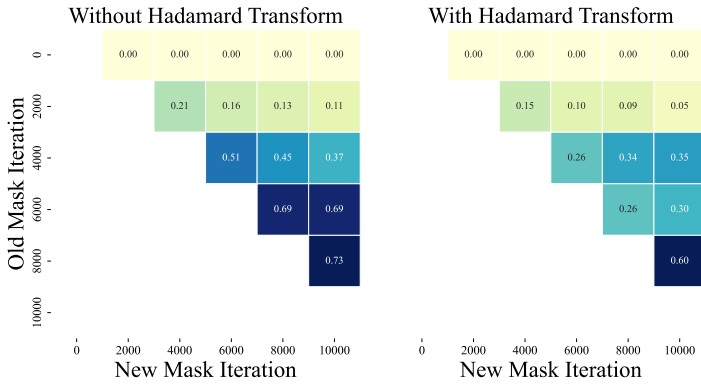

*Figure 8.* Fraction of masked values retained from an old iteration to a new iteration for a 30M model trained with QuEST W8A8.

## A.2. The 1-bit Case

To determine the optimal outer trust scaling factor $s^*$, discussed in Section 3.3, we conduct a sweep over $s$, varying the outer size of the outermost trust regions as $T = s \cdot \frac{\alpha^*}{2^b - 1}$. The results for 1-bit, shown in Figure 9, indicate that $s^* = 1.30$ for the standard QuEST setup and $s^* = 1.25$ for the setup without the Hadamard Transform (HT), corresponding to exactly a quarter of the quantization interval.

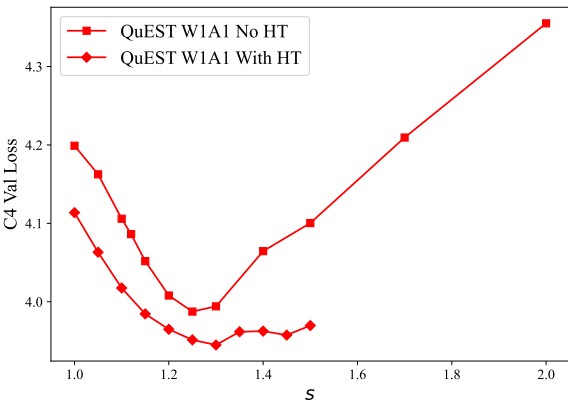

*Figure 9.* Performance of QuEST as a function of the outer trust scaling factor $s$ for a 30M model pretraining.

### A.3. Zero-shot Evaluation of QuEST Models

To assess the effectiveness of QuEST beyond perplexity, we conducted a comprehensive zero-shot evaluation on five established commonsense reasoning benchmarks: HellaSWAG (Zellers et al., 2019), ARC (Easy and Challenge) (Clark et al., 2018), PiQA (Bisk et al., 2019), and Winogrande (Sakaguchi et al., 2019). We compared multiple QuEST quantization settings against full-precision (BF16) baselines. All models were trained on 80B tokens unless otherwise noted.

Table 3 summarizes the zero-shot accuracy across these tasks. Overall, W4A4 QuEST closely matches its BF16 counterpart on HellaSWAG and PiQA, with minor degradation on ARC and Winogrande. Sparse quantization ("2:4 INT4") incurs larger drops.

| Method, Model Size | HSWAG (%) ↑ | ARC-e (%) ↑ | ARC-c (%) ↑ | PiQA (%) ↑ | Winogrande (%) ↑ |
|---|---|---|---|---|---|
| BF16, 800M | 39.51 | 53.28 | 22.44 | 71.65 | 53.91 |
| QuEST INT4, 800M | 39.18 | 52.40 | 22.01 | 71.16 | 52.96 |
| QuEST no HT INT4, 800M | 38.03 | 52.44 | 22.70 | 71.11 | 51.38 |
| QuEST 2:4 INT4, 800M | 36.26 | 50.46 | 21.08 | 69.04 | 53.75 |

*Table 3.* Zero-shot evaluation on five commonsense reasoning benchmarks.

## B. Additional Information about the Experimental Setup

### B.1. Model Hyper-parameters

For our experiments, we chose to use the Llama 2 (Touvron et al., 2023) model as the base architecture. For the attention block, this architecture utilizes multi-head attention (Vaswani et al., 2023) with rotary positional embeddings (Su et al., 2023). For the MLP block, it uses additional gate projection and SiLU (Elfwing et al., 2017) activation function. We kept the MLP intermediate dimension equal to $8/3$ of the hidden size, padding it to 256 for increased kernel compatibility. For the AdamW optimizer, we used $\beta_1 = 0.90$ and $\beta_2 = 0.95$. We did not apply weight decay to any biases and layer normalizations. Table 4 describes size-specific models and optimizer hyper-parameters for all model sizes used in this work.

| Model size | 30M | 50M | 100M | 200M | 430M | 800M |
|---|---|---|---|---|---|---|
| Num. Blocks | 6 | 7 | 8 | 10 | 13 | 16 |
| Hidden Size | 640 | 768 | 1024 | 1280 | 1664 | 2048 |
| Num. Attn. Heads | 5 | 6 | 8 | 10 | 13 | 16 |
| Learning Rate | 0.0012 | 0.0012 | 0.0006 | 0.0003 | 0.00015 | 0.000075 |
| Num. Tokens | 3B | 5B | 10B | 20B | 43B | 80B |

*Table 4.* Hyper-parameters used for each model size.

## B.2. Training Stability and Convergence

Here we present the loss curves for BF16, LSQ, PACT, and QuEST (ours) to analyze training stability and convergence. As shown in Figure 10(a), QuEST smoothly converges throughout training, closely tracking the BF16 baseline while consistently outperforming LSQ. Meanwhile, PACT struggles with much higher loss, indicating poor convergence. To better highlight the differences between QuEST and LSQ in the later stages of training, Figure 10(b) focuses on steps after 1000, removing PACT for clarity. This zoomed-in view shows that QuEST maintains a consistently lower loss trajectory than LSQ, further reinforcing its superior stability and accuracy across training.

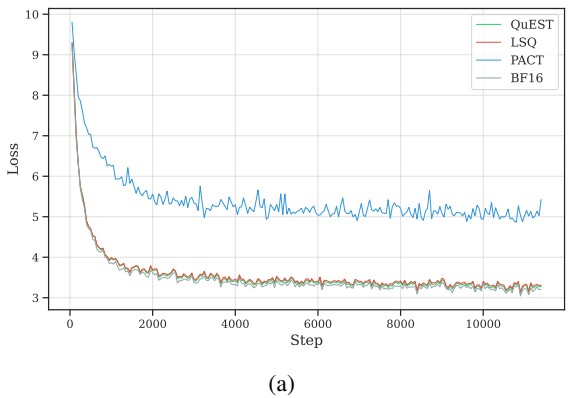
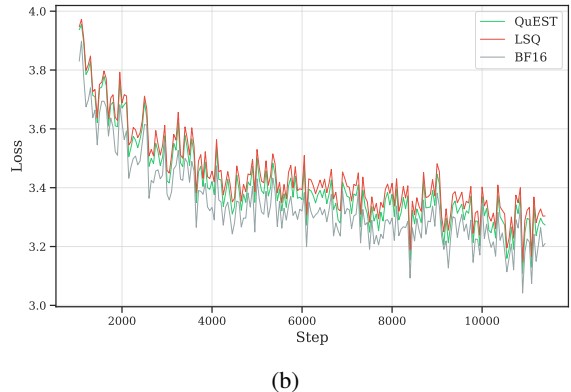

(a)                                                   (b)

*Figure 10.* Training loss curves for a 30M model trained on 3B tokens with W4A4 bitwidth, comparing QuEST (ours), LSQ, PACT, and BF16. **(a)** Full training loss curves, showing that QuEST closely follows BF16 and consistently outperforms LSQ, while PACT struggles with high loss. **(b)** Zoomed-in view of training steps after 1000, excluding PACT for clarity, highlighting that QuEST maintains a lower loss than LSQ throughout training.

## B.3. Hyper-parameter Search for Baseline Methods

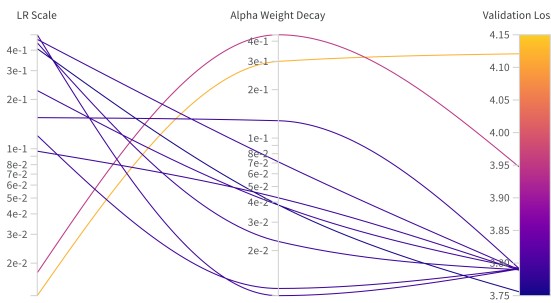

*Figure 11.* Hyperparameter search for PACT on a 30M parameter model with 4-bit weights and activations, trained on 10% of the dataset. The search explores different values for learning rate scaling (LR Scale) and alpha weight decay, with validation loss indicated by the color gradient. Lower validation loss (darker colors) corresponds to better configurations.

To ensure fair comparisons between QuEST and prior QAT methods, we conducted hyperparameter searches for both PACT and LSQ. Given PACT's instability at lower bitwidths, we extensively tuned two key hyperparameters: weight decay and learning rate scaling $s$ for the quantization parameter $\alpha$ (i.e., $\eta_\alpha = s \times \eta$). Figure 11 shows the loss achieved across different weight decay and LR scale values.

For LSQ, we only tuned weight decay, as the LSQ formulation already applies scaling internally to the gradient of $\alpha$, making additional learning rate adjustments unnecessary. Table 5 summarizes the results of the weight decay search across 2-bit, 3-bit, and 4-bit LSQ models, where the best-performing configuration (highlighted in bold) was used for final model comparisons.

| Weight Decay | 2-bit PPL ↓ | 3-bit PPL ↓ | 4-bit PPL ↓ |
|---|---|---|---|
| 0.001 | 37.02 | 31.10 | 27.93 |
| 0.01 | 36.91 | 30.89 | 27.72 |
| 0.1 | **36.54** | **30.26** | **27.51** |
| 1.0 | 38.12 | 31.16 | 28.67 |

*Table 5.* Weight decay hyperparameter search results for LSQ across different bitwidths of 30M model. The best-performing setting is highlighted in bold.

Our hyperparameter search ensured that LSQ and PACT were tuned optimally before comparing against QuEST, leading to a fair evaluation of performance across all tested quantization methods.

## C. Scaling Laws

### C.1. Description of the Fitting Procedure

As described in Section 4.3, we closely follow the fitting procedure of Hoffmann et al. (2022) for the scaling law (5) fitting. Specifically, we copied their grid of initialization given by: $\alpha \in \{0., 0.5, \dots, 2.\}$, $\beta \in \{0., 0.5, \dots, 2.\}$, $e \in \{-1., -.5, \dots, 1.\}$, $a \in \{0, 5, \dots, 25\}$, and $b \in \{0, 5, \dots, 25\}$. We also reuse their $\delta = 10^{-3}$ for the Huber loss. In addition, we fit the eff($P$) coefficient for a number of quantization schemes described below:

- QuEST for $P \in \{1, 2, 3, 4, 8\}$.
- Weight-only QuEST for $P \in \{1, 2, 3, 4\}$.
- QuEST without the HT for $P \in \{1, 2, 3, 4, 8\}$.
- QuEST with FP4 grid.
- QuEST with 2:4 INT4.

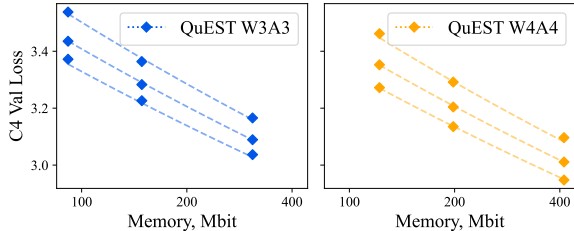

*Figure 12.* Scaling law (5) fit for 3 and 4 bit QuEST with tokens/parameters ratios in $\{25, 50, 100\}$.

### C.2. Analysis of the Transitory Data Regime

The results in Section 4.4 suggest that 4-bit training is optimal in the $D/N \to \infty$ regime. Here, we use the fitted scaling law (5) to verify that 4 bit is also close to optimal for $D/N$ ratios that are reasonable in practice. We formulate the question as follows: for a fixed model size (e.g. in Gb), for which amount of compute is QuEST 4-bit the optimal precision?

Figure 14 demonstrates the (predicted) dependence of performance as a function of $\frac{D}{N} \cdot \frac{16^2}{P^2}$. For BF16, this quantity becomes $D/N$. For other $P$, it ensures the same amount of training computed ($\sim ND$). As such, models there are compared at both the same size and the same training compute. We can see that 4-bit quantization becomes optimal after it passes a certain compute threshold that depends on model size. We can also see that the threshold value decreases as the model size (in Gb) grows. For a 14.0Gb model (corresponding to 7B parameters in BF16), the threshold is around $D/N \approx 30$, which is significantly below the amount of data that models of that size are currently trained on (see Section 4.3). For even larger models, the threshold eventually becomes less than the "Chinchilla-optimal" ratio of $D/N \approx 20$. This validates that the regime in which 4-bit pre-

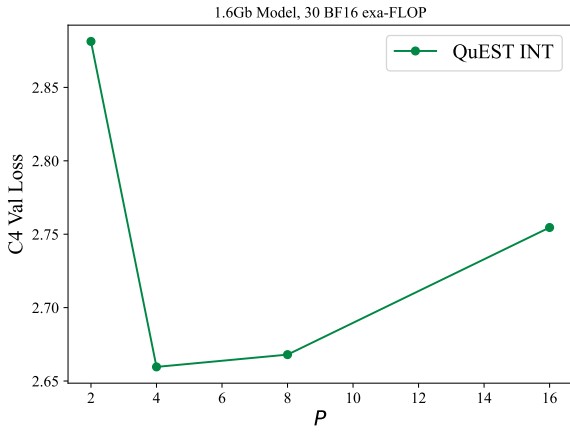

*Figure 13.* Comparison of different QuEST precisions $P$ at a fixed model size and training compute.

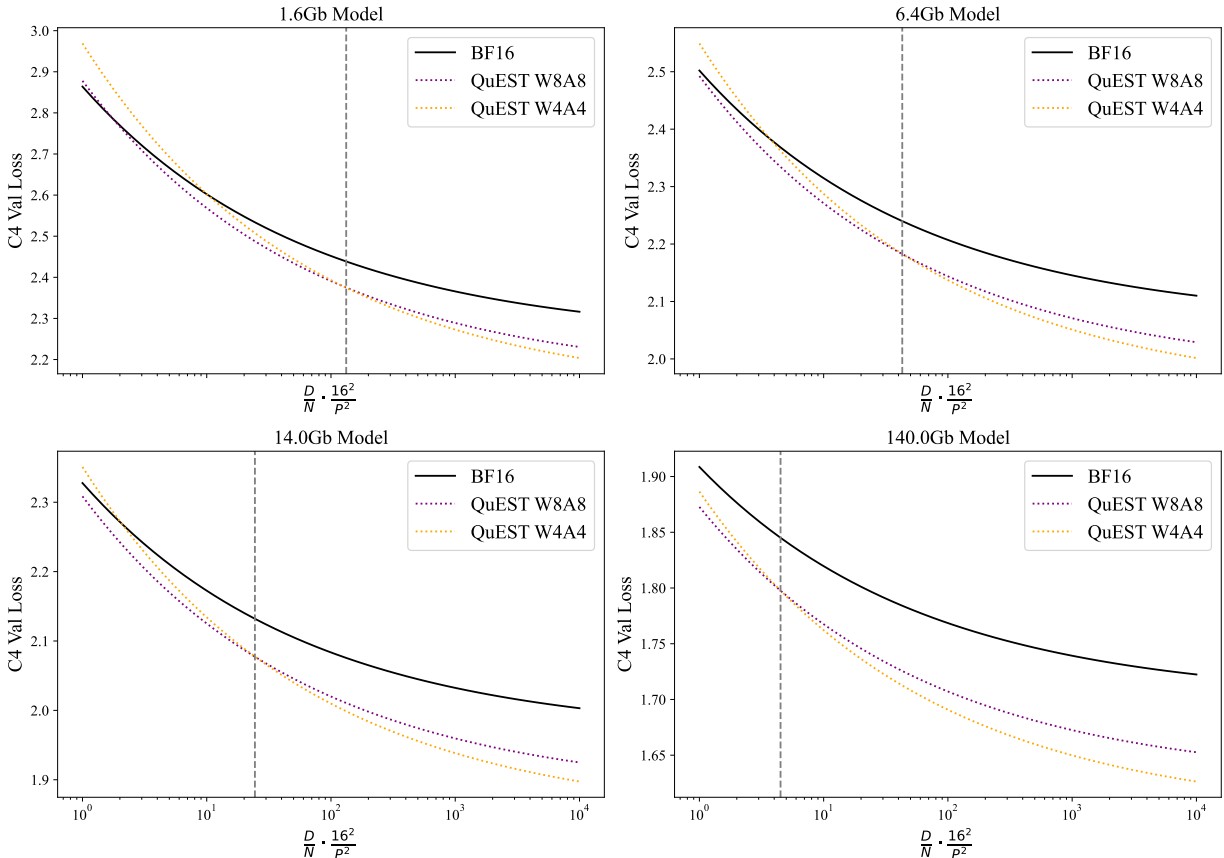

*Figure 14.* Different QuEST precision performance as a function of tokens-to-parameters ratio at a fixed model memory footprint. The gray line indicates a 4-bit optimality threshold.

training is optimal can, in fact, be easily achieved in practice.

We validate this in practice by training a set of models of approximately the same model size (1.6 Gb) and training compute (30 exa-FLOP, 100B tokens for BF16 100M). The results, presented on Figure 13, show how $P = 4$ is optimal.

