# OpenReview forum: "QuEST: Stable Training of LLMs with 1-Bit Weights and Activations"
_ICML.cc/2025/Conference — ICML 2025 poster_

### Official Review · Reviewer_Xwq9 · 2025-03-08

**Overall Recommendation:** 2

**Summary:**

The paper presents QuEST, a quantization-aware training (QAT) method that enables training LLMs with extremely low-precision weights and activations (both down to 1-bit). The key contributions are: (1) Hadamard normalization with MSE-optimal fitting for quantization, and (2) a trust gradient estimator that minimizes the difference between quantized and full-precision gradients. The method claims to achieve Pareto-competitive performance with FP16 while using 4-bit or lower precision.

## update after rebuttal

After reviewing the authors' rebuttal, I maintain my weak reject recommendation. While the authors made efforts to address several concerns, limitations remain:

1. Scale and Evaluation:
- The authors' testing remains limited to relatively small models (up to 800M parameters, with a mentioned 1.2B test). While they claim to be working on 3B, this still falls short of demonstrating effectiveness at production-relevant scales (7B+).
- The authors did not address my concern about performance on reasoning tasks like GSM8k or AIME, only showing basic zero-shot evaluations on simpler tasks like HellaSwag.

2. QAT vs PTQ Comparison:
- The authors' comparison with PTQ methods at small scales (800M) is methodologically questionable, as PTQ methods typically show better performance with increased model scale due to greater parameter redundancy.
- While they provided some results from QuaRot at 70B scale, this comparison doesn't fully address the fundamental differences between QAT and PTQ at scale.
- Most concerning is the computational cost: modern PTQ methods can quantize 70B models in about an hour on a single A100, while the authors quote 1600 A100 hours for just training an 800M model with QuEST, and estimate 40,000 hours for a 3B model. This huge difference in resource requirements makes QuEST far less competitive compared to a PTQ+SFT flow in practical scenarios.

3. Technical Novelty:
- The use of Hadamard Transform (HT) for improving gradient flow isn't as novel as presented - though not directly equivalent, similar techniques appeared in FAT from 4 years ago.
- QuaRot, a well known PTQ scheme, also uses Hadamard transform in both weights and activations. Thus I won't see Hadamard transform as a fundamental novelty in QuEST.

4. Broader Impact:
- Without demonstrating effectiveness on larger models and more challenging tasks (especially reasoning tasks), the practical impact of this work remains uncertain.
- The authors' rebuttal suggests that reasoning tasks are "outside the reach of 1B-parameter models without explicit instruction fine-tuning," but this sidesteps the important question of how their method performs after fine-tuning.

While the paper presents some interesting technical contributions, these fundamental limitations in scale, evaluation breadth, novelty, and especially the prohibitive computational requirements make it difficult to recommend for acceptance at ICML. The work would benefit from more comprehensive large-scale evaluations, clearer differentiation from existing approaches, and better justification for its high computational costs versus existing PTQ solutions.

## Additional post-rebuttal comments

After further examining the materials, I would like to bring up two further issues which I believe will become viewable to the authors for improving their work:

### 1. Mischaracterization of FAT vs. QuEST

The authors claim their use of Hadamard Transform (HT) is new & fundamentally different from FAT, but this is not accurate. While there are implementation differences, both methods transform weight representations through frequency domains to improve quantization:

- In FAT (Fig. 3 and Supplementary Section 2.2), gradient flow explicitly passes through the Fourier domain during backpropagation with ∂Wt/∂W being a function of frequency components
- Similarly, QuEST employs the Hadamard domain for gradient flow, and their "trust estimator" also operates in this transformed space
- Both approaches use masking technique and both aim to achieve the same goal: improving gradient estimation by leveraging frequency-based transformations

The core innovation of "transforming to a more quantization-friendly domain" is shared between these approaches, though applied in different contexts.

### 2. Critical limitations in practical applicability

The authors are encouraged to check out the recent "Quantization Hurts Reasoning" paper (https://arxiv.org/abs/2504.04823) which demonstrates why evaluating reasoning capability at low bitwidth is crucial for modern LLMs. This paper shows:

- Severe degradation in complex reasoning tasks (e.g., AIME) when quantizing below 8 bits
- Larger models (32B-70B) fare significantly better than smaller ones under aggressive quantization
- Models' origins (distilled vs. RL-trained) substantially impact quantization tolerance

QuEST's failure to demonstrate effectiveness beyond toy-scale models (800M-1.2B) leaves its practicality questionable for the most important use case: enabling efficient inference of large reasoning models (7B+). The computational cost of QuEST training (40,000 GPU hours estimated for just a 3B model) presents a prohibitive barrier to practical adoption.

This makes QuEST's contribution largely theoretical rather than practical, especially when PTQ methods can quantize 70B models in hours (my experience is <1hr) with acceptable reasoning performance degradation.

**Claims And Evidence:**

The claims are partially supported by evidence, but with several limitations:

- The experiments only cover relatively small models (30M-800M parameters). The authors didn't provide justification of not going to higher sizes.
- Testing is limited to C4 dataset only, making all conclusions restrictive, e.g., lacking reasoning tasks like GSM8k or AIME performance
- Improvements over baselines (PACT, LSQ) are modest (e.g., ~2.6 point improvement in W1A1 configuration)
- The paper claims GPU speedups for 7B models but only projects them without actual training

**Essential References Not Discussed:**

Several critical missing references and comparisons:

1. Recent theoretical works on precision:
- "Scaling Laws for Precision" (https://arxiv.org/abs/2411.04330) - Shows fundamental relationships between model precision and performance
- "How Numerical Precision Affects Mathematical Reasoning Capabilities of LLMs" (https://arxiv.org/abs/2410.13857) - Demonstrates theoretical limitations of low-precision training

2. Recent PTQ advances:
- "ARB-LLM: Alternating Refined Binarizations for Large Language Models" (https://github.com/zhitengli/arb-llm) - Achieves strong results with binary weights
- "QuaRot: Outlier-Free 4-Bit Inference in Rotated LLMs" (https://arxiv.org/abs/2404.00456) - Successfully applies Hadamard transform in PTQ setting

The omission of these works is particularly concerning because:

- The use of Hadamard transform (HT) in QuEST is presented as novel, but QuaRot has already demonstrated its effectiveness in the more challenging PTQ setting, achieving strong results on much larger models (LLaMA2-70B). Their success in PTQ makes HT's effectiveness in QAT unsurprising.
- Recent developments in HT optimization (as shown in PyTorch's HadaCore) demonstrate that the technique is becoming standard in quantization workflows. The paper should acknowledge this broader context.
- The theoretical framework from Kumar et al. and Feng et al. provides important context about fundamental precision-performance trade-offs that should inform any QAT method.

**Experimental Designs Or Analyses:**

Experimental results appear valid. Can you explain why in Fig. 1 the W3A3 and W4A4 curves overlap?

**Methods And Evaluation Criteria:**

The methods are sound but narrow in scope:

- Use of C4 dataset alone is limiting
- Model sizes tested (up to 800M) are not aligned with current community standards
- Baseline comparisons (PACT, LSQ from 2018-2020) are quite outdated
- Primary metrics (perplexity, validation loss) need broader validation

**Other Comments Or Suggestions:**

- Editorial issues (repeated definitions of QAT, STE, etc., typos like "Gaussiant")

- While the paper presents interesting ideas and achieves stable low-precision training, the limited scope of evaluation, modest improvements, and several important missing comparisons suggest the work needs substantial revision.

**Other Strengths And Weaknesses:**

Strengths:

- Novel integration of Hadamard Transform in QAT
- Stable training achieved at extreme low precision (1-bit)
- Practical GPU implementation with demonstrated speedups
- Clear theoretical framework for trust estimation

Weaknesses:

- Limited model scale (only up to 800M parameters)
- Narrow experimental validation (single dataset)
- Marginal improvements over baselines
- Some figures need better explanation (e.g., overlapping W3A3 and W4A4 curves in Figure 1)

**Questions For Authors:**

QUESTIONS FOR AUTHORS:

- Why is the method named QuEST?
- Can you explain the overlapping W3A3 and W4A4 curves in Figure 1?
- What challenges do you anticipate in scaling to larger models (7B+)?
- How does your method compare with recent works like ARB-LLM and QuaRot?

**Relation To Broader Scientific Literature:**

Several critical omissions in literature discussion:

- Should discuss relationship to recent work on precision scaling laws (Kumar et al., 2024)
- Should compare against recent work on numerical precision effects (Feng et al., 2024)
- Missing comparison with state-of-the-art PTQ work like ARB-LLM
- The meaning of "QuEST" is never explained

See below for more details.

**Theoretical Claims:**

The theoretical framework appears sound, but several clarifications are needed:

- The derivation of trust estimation is mathematically correct
- The paper should clarify if α in equation (1) is per-output-channel scaling
- The relationship between HT and gradient estimation could be better explained

---

> ### Author Rebuttal · Authors · 2025-04-01
>
> > Why is the method named QuEST?
>
> We apologize for the omission, QuEST stands for **Qu**antized, **E**fficient, and **S**table **T**raining.
>
> > Can you explain the overlapping W3A3 and W4A4 curves in Figure 1?
>
> The decrease in training precision corresponds to fixed parameter count models shifting to the left. The performance degradation due to increased compression increases the loss and shifts the points upward. Between W3A3 and W4A4, those effects cancel out, landing the points on seemingly “the same” loss-to-model-size curve. In Section 4.4, we presented the “precision efficiency (eff(P))” metric to quantify this trade-off and accurately determine the optimal precision to be W4A4.
>
> > What challenges do you anticipate in scaling to larger models (7B+)?
>
> We did not observe any training instabilities or unexpected performance degradation when scaling the model size. As such, the main challenge we anticipate is the need to commit hundreds of thousands of GPU-hours to test this novel approach for a production-scale pre-training run.
>
> > Comparison with recent works like ARB-LLM, QuaRot, and Kumar et al. (2024).
>
> We did not include a comparison to PTQ methods (QuaRot or ARB-LLM) since they are not competitive with QAT: many PTQ methods work in one-shot, whereas QAT methods perform training or re-training by definition.
>
> To fully address this concern, below we present a comparison of C4 validation PPL of the 800M models trained by us using BF16, QuEST INT4, and QuEST INT8. We then show numbers obtained relative to QuaRot PTQ (as suggested by the reviewer), and round-to-nearest quantization (RTN):
>
>
> | BF16  | QuEST W4A4 | RTN W4A4 | QuaRot W4A4 | RTN W8A8 | QuaRot W8A8 |
> |------|------------|-----------|-------------|-----------|------------|
> | 11.72  | 12.12               | 53.73         | 46.85               | 12.60          | 12.59               |
>
> These results clearly show that PTQ isn’t competitive with QAT. Specifically, notice that our W4A4 model is _more accurate_ than the QuaRot model in W8A8.
> To your question, ARB-LLM focuses on the simpler problem of weight-only quantization which is not the main focus of our work, whereas Feng et al. (2024) focuses on PTQ. As such, we largely omitted these comparisons from the paper.
>
> We will provide additional background citations and clarification on this topic in the next version of the paper.
>
> Please see the answer to [Reviewer HH36](https://openreview.net/forum?id=I0Ux2nAN6u&noteId=h10Gvw440m) for the relationship with Kumar et al. (2024).
>
> > Experiments only cover small models (30M-800M parameters).
>
> Notably, the largest models we trained (800M) closely match the size of the _Meta Llama-3.2-1B_ model (up to a smaller embedding layer). This puts us at the lower end of current standards in terms of model sizes. Importantly, our study considers slightly larger models than the concurrent work of Kumar et al. (2024).
>
> Runs on the 800M model require around \~1600 A100 GPU hours, with the total experimental cost of our submission being of around 12000 GPU hours. This makes it hard to scale further in an academic setup. Nevertheless, we have confirmed out results at 1.2B-parameter scale in the answer to [Reviewer HH36](https://openreview.net/forum?id=I0Ux2nAN6u&noteId=h10Gvw440m), and are working towards a 3B-parameter run on 300B tokens (\~40,000 A100 GPU hours).
>
> We hope the reviewer can appreciate that this is very computationally-intensive and does not fit within the scope of the rebuttal.
>
> > Testing is limited to C4 dataset only, making all conclusions restrictive, e.g., lacking reasoning tasks like GSM8k or AIME performance.
>
> Please note that our testing is not limited to C4: Appendix A.3 included the 0-shot evaluations of some of the models. To better address this issue, we present additional 0-shot evaluations (HellaSwag, ARC-Challenge, ARC-Easy, PiQA, Winogrande) of a broader set of models we trained:
>
> https://github.com/QuEST2025/speedup/blob/main/zero-shots.md
>
> These results are again consistent with the C4 evaluations. As for the reasoning tasks like GSM8k or AIME, they are outside the reach of 1B-parameter models without an explicit instruction fine-tuning phase.
>
> > The success of the Hadamard transform in PTQ makes HT's effectiveness in QAT unsurprising.
>
> **Regarding the Hadamard Transform (HT):** We emphasize that the context in which we analyze HT is different from all the mentioned PTQ methods. Specifically, PTQ methods utilize the Hadamard transform to mitigate outliers in model weights and activations, and obtain better quantization grid fit.
> In addition to this, we show novel effects of HT on:
>  1. Improving gradient estimator alignment, as discussed in Section 3.2 (line 205)
>  2. Circumventing the “dead weight problem” in gradient masking, as discussed in Section 3.3 (line 215) and Appendix A.1, by making sure that all weights get gradient.
>
> These effects are unique to QAT; to the best of our knowledge, we are the first to explore them.

---

> > ### Comment · Reviewer_Xwq9 · 2025-04-02
> >
> > While I appreciate the authors' response, most concerns are not quite addressed. For one thing, comparing QAT & PTQ in the small model size regime is unfair, as PTQ performs better & better as model size scales up with more parameter redundancy and higher tolerance to quantization. Second, using frequency-domain transform (HT practically does this) for better gradient flow isn't new, e,g,. "FAT: Learning Low-Bitwidth Parametric Representation via Frequency-Aware Transformation" from 4 years back was already doing this, though not in the context of LLM.

---

> > > ### Author Response · Authors · 2025-04-03
> > >
> > > Thank you for the opportunity to address your remaining concerns.
> > >
> > > > 1. Comparing QAT & PTQ in the small model size regime is unfair, as PTQ performs better & better as model size scales up.
> > >
> > > We apologize for the misunderstanding: we interpreted your first request as asking us to directly compare PTQ with QAT in our setting, and since the Kumar et al. reference you pointed to does in fact do this: they apply PTQ to models from 30M and 220M in their Figure 1.
> > > We do agree that this comparison is not  very meaningful on small models.
> > >
> > > To address the substance of your question, we examine the performance of state-of-the-art weights and activations (W&A) PTQ methods **at scale**, and compare it with QuEST:
> > >
> > > - The state-of-the-art PTQ methods are QuaRot (NeurIPS24), and SpinQuant (to appear in ICLR25). Focusing on SpinQuant, we observe that 4-bit W&A quantization is still far from lossless, even at 70B scale. Please see Table 5 at https://arxiv.org/pdf/2405.16406, showing a significant 4.4 avg. 0-shot drop for Llama-3 70B. We would expect 2-bits or below to provide terrible recovery with PTQ. We believe that the key compression difficulty addressed by QAT is the quantization of model activations, containing massive outliers in LLMs.
> > >
> > > - Broadly, the results of Kumar et al., reproduced by https://arxiv.org/pdf/2411.17691, suggest that PTQ methods become _worse_ as the training tokens increase. QAT is not affected by this; in fact, our scaling laws suggest that QAT becomes better as toks/params increase (see Appendix C.2).
> > >
> > > We hope this addresses your concern. We include further relevant comparisons on point 3 below.
> > >
> > > > 2. Second, using frequency-domain transform for better gradient flow isn't new, e,g,. FAT is already doing this.
> > >
> > > We thank you for raising this interesting reference. Having examined it thoroughly, we respectfully point out that there are major differences between our results, what the authors propose, and your characterization of their results:
> > >
> > > 1. First, the FAT approach is different from ours: please see their Figure 3 and Sec 3.3. What they do is **a) transform the weights via DCT, b) perform a parametrized filtering over the weights, c) transform back the filtered weights, and then finally d) quantize the weights in the standard domain**.  At inference time, the filtering and transform components are dropped from the model. Moreover, they do not perform any kind of filtering over activations, as this would be prohibitive at runtime.
> > >
> > > 2. By contrast, in QuEST: we a) transform **both weights and activations** into Hadamard domain; b) we perform distribution-matching, clipping, and quantization **for both weights and activations, in the Hadamard domain**; c) **the gradient flow “switches” between domains**.
> > >
> > > We hope it is clear that the two approaches are different: FAT is clearly developed with CNN filters in mind, and it is not obvious to us how it would be applied to LLMs. **Irrespective of this, we hope the reviewer can agree that FAT does not address activation quantization, critical in LLMs, at all: they simply do RTN on activations.**
> > >
> > > 3. Finally, we try to address a key concern from the original review:
> > >
> > > > modest improvements, and missing comparisons
> > >
> > > We performed a comparison between QuEST and:
> > > * **STE** (BNN, Courbariaux et al.)
> > > * Hadamard + STE, i.e. **QuaRot** with a backward pass (but without our fitting and trust factors)
> > > * Activation-only quantization via STE (**AO STE**), which is an extremely generous upper bound on the performance of _FAT_: we have zero error  _on the weights_, and apply STE on the activations, as they do.
> > >
> > > The results are provided below, for 30M and 50M models.
> > >
> > > | Model size | Method    | W4A4  | W3A3  | W2A2  | W1A1  |
> > > |-|-|-|-|-|-|
> > > | 30M  | STE       | 3.792 | 4.449 | 4.793 | 5.256 |
> > > | 30M   | AO STE    | 3.658 | 4.181 | 4.549 | 5.004 |
> > > | 30M   | QuaRot    | 3.338 | 3.612 | 4.481 | 4.932 |
> > > | 30M   | QuEST     | 3.272 | 3.372 | 3.574 | 3.945 |
> > > | 50M   | STE       | 4.040 | 4.542 | 5.162 | 6.867 |
> > > | 50M    | AO STE    | 3.733 | 4.315 | 4.601 | 4.985 |
> > > | 50M        | QuaRot    | 3.201 | 3.695 | 4.566 | 5.007 |
> > > | 50M        | QuEST     | 3.135 | 3.226 | 3.441 | 3.791 |
> > > C4 Val Loss, D/N=100
> > >
> > > To more precisely gauge **the scale of improvements** we introduce a **iso-loss size improvement** metric.  A better QAT method will need a smaller model size to reach the same loss. For each QAT method X, from its scaling law we compute the size of the model we would need to train **using QuEST** in order to match the same loss as X.
> > >
> > > |bitwidth|Eff(P)||||QuEST Size Reduction|||
> > > |-|-|-|-|-|-|-|-|
> > > ||QuEST|LSQ|QuaRot|AO STE|Over LSQ|Over QuaRot|Over AO STE|
> > > |W4A4|0.69|0.56|0.48|0.09|19%|31%|87%|
> > > |W3A3|0.43|0.32|0.11|0.01|25%|74%|98%|
> > > |W2A2|0.15|0.12|0.00|0.00|21%|98%|99%|
> > >
> > > We believe this makes it evident that 1) QuEST is significantly superior to prior methods, and 2) for QuaRot and STE, this advantage increases as we decrease target precision.

---

### Official Review · Reviewer_HH36 · 2025-03-12

**Overall Recommendation:** 2

**Summary:**

In this paper, the authors propose QuEST, a low-bit quantization-aware training (QAT) method aimed at compressing models more accurately. Experiments demonstrate that QuEST outperforms the LSQ method in performance under various low-bit weight and activation quantization scenarios. Additionally, based on the designed INT4 kernel, the Linear computation per layer shows a better speedup ratio compared to BF16.

**Claims And Evidence:**

Yes.

**Essential References Not Discussed:**

None

**Experimental Designs Or Analyses:**

Yes. The experimental design is not very reasonable.
In the speedup ratio experiments, the paper only compares QuEST with BF16, which is insufficient to comprehensively evaluate the effectiveness of the method. Additionally, tests should be conducted with different sequence lengths and batch sizes to observe the performance variations of QuEST under different scenarios. The current experimental analysis remains incomplete, and more comparative experiments are needed to support its efficiency claims.

**Methods And Evaluation Criteria:**

Yes.

**Other Comments Or Suggestions:**

None

**Other Strengths And Weaknesses:**

1. In Section 4.3, Scaling Laws are mentioned, and related experiments are conducted in Section 4.6. The stable Scaling Laws indicate that the QuEST method exhibits strong generalization capabilities across models of different scales, meaning its performance improvement patterns remain consistent across LLMs of varying sizes. This experiment may suggest that QuEST can still work effectively on larger models, but the paper does not provide a detailed discussion on how its advantages manifest in such scenarios.
2. In Section 5, Kernel Overview, the paper mentions that the third stage utilizes an enhanced CUTLASS and optimizes GEMM operations, but it does not elaborate on the specific implementation. From an optimization perspective, this may involve the following aspects:
GEMM kernel optimization: Potentially using more efficient INT4 computation layouts to improve Tensor Core utilization.
Memory access optimization: Possibly reducing data movement and optimizing shared memory loading to enhance throughput.
Parallel computation optimization: Potentially improving warp-level scheduling or data flow optimization to boost computational efficiency. However, the paper does not provide detailed explanations of these optimizations, which need further clarification.
3. In the speedup ratio experiments, the paper only compares QuEST with BF16, which is insufficient to comprehensively evaluate the effectiveness of the method. Additionally, tests should be conducted with different sequence lengths and batch sizes to observe the performance variations of QuEST under different scenarios. The current experimental analysis remains incomplete, and more comparative experiments are needed to support its efficiency claims.

**Questions For Authors:**

1. In Section 4.3, Scaling Laws are mentioned, and related experiments are conducted in Section 4.6. The stable Scaling Laws indicate that the QuEST method exhibits strong generalization capabilities across models of different scales, meaning its performance improvement patterns remain consistent across LLMs of varying sizes. This experiment may suggest that QuEST can still work effectively on larger models, but the paper does not provide a detailed discussion on how its advantages manifest in such scenarios.
2. In Section 5, Kernel Overview, the paper mentions that the third stage utilizes an enhanced CUTLASS and optimizes GEMM operations, but it does not elaborate on the specific implementation. From an optimization perspective, this may involve the following aspects:
GEMM kernel optimization: Potentially using more efficient INT4 computation layouts to improve Tensor Core utilization.
Memory access optimization: Possibly reducing data movement and optimizing shared memory loading to enhance throughput.
Parallel computation optimization: Potentially improving warp-level scheduling or data flow optimization to boost computational efficiency. However, the paper does not provide detailed explanations of these optimizations, which need further clarification.
3. In the speedup ratio experiments, the paper only compares QuEST with BF16, which is insufficient to comprehensively evaluate the effectiveness of the method. Additionally, tests should be conducted with different sequence lengths and batch sizes to observe the performance variations of QuEST under different scenarios. The current experimental analysis remains incomplete, and more comparative experiments are needed to support its efficiency claims.

**Relation To Broader Scientific Literature:**

None

**Theoretical Claims:**

Yes.

---

> ### Author Rebuttal · Authors · 2025-04-01
>
> > Detailed discussion on how its (QuEST’s) advantages manifest.
>
> The fact that QuEST enables stable training across scales has the following implications:
>
> 1. If using QuEST, INT4 is the “optimal” bit-width for weights and activations in terms of inference effectiveness, that is, the accuracy that can be obtained at a given model size. This is a strict improvement relative to STE, which was found in Kumar et al. (2024) and in our experiments to only provide competitive low-bit training at 7-8 bit weights and activations.
>
> 2. As the reviewer remarked, this finding holds and transfers across all model scales: thus, future large-scale pre-training runs could leverage this technique to produce highly-accurate models with low precision.
>
> 3. Our approach also enables a direct comparison between the “effectiveness” of different precisions P, namely the efficiency factor eff(P). Thus, based on runs on small models, the user can determine the “optimal” precision for a given model architecture and hardware target (which may influence the set of precisions supported).
>
> To validate our findings at a larger scale, we trained a 1.2B-parameter model over 40B tokens (2’000 A100 GPU hours), which was the largest size we could train on our academic cluster. The accuracy findings, including C4 loss and 0-shot evaluations, confirm that our findings indeed scale to this larger model size: https://github.com/QuEST2025/speedup/blob/main/1200M.md
>
> We are currently working towards a 3B-parameter run on 300B tokens (~40’000 GPU hours), but we hope the reviewer can appreciate that this is very computationally-intensive and does not fit within the scope of the rebuttal.
>
> > Detailed explanations of [GPU Kernel] optimizations.
>
> We believe there may be a slight confusion here. As stated in the paper, we utilize the highly optimized CUTLASS operations for the “raw” matrix multiplications in both 16-bit (BF16) and 4-bit (INT4) precisions. Thus, these basic operations are heavily optimized by NVIDIA, the makers of the library and the hardware.  Our main focus is to reduce the inference overheads over the QuEST format: that is, quantization/dequantization, clipping and Hadamard multiplication. This is overviewed in Section 5, and we would be happy to describe it further in the discussion.
>
> > The paper only compares QuEST with BF16, which is insufficient to comprehensively evaluate the effectiveness of the method.
>
> Our speedup ratio experiments compare: the BF16 baseline, our approach (QuEST), and an idealized 4bit version which does not require the Hadamard multiplication (No HT).
>
> We use BF16 data type in the baseline because
>
> 1. Our experiments use smaller-scale Llama-3 models. The full-precision data type for Llama-3 models is BF16.
>
> 1. BF16 and FP16 are common weight types in the popular open-source models (e.g., Llama, Qwen, etc.). BF16 and FP16 are equally supported by modern GPUs (e.g., Ampere and Hopper architectures). They have the same computational performance. The results for BF16 also apply to FP16.
>
> Our BF16 baseline is the official NVIDIA libraries (i.e., cuBLAS/cuDNN) that implement the GEMM routine used under-the-hood by e.g. PyTorch. These codes have been heavily optimized by NVIDIA to achieve near-optimal performance on their hardware.
>
> The results show that:
>
> 1. Our approach leads to significant inference speedups relative to the extremely competitive full-precision baseline, which is standard for all open models, especially on large matrices. Note that this occurs at the same or better accuracy (as per our results).
>
> 1. The overheads of our format, including activation clipping and Hadamard transforms, are small (less than 10% on average, and 30% in the worst case).
>
> 1. Our results are not surprising, since they track well with the expected speedups from low-precision GEMM for the corresponding matrix sizes.
>
> We would be happy to run additional comparisons that the reviewer would find relevant in this context.
>
> > Tests should be conducted with different sequence lengths and batch sizes.
>
> We agree; to address this, we conducted more tests on different sequence lengths and batch sizes. You can find the speedup results using this link https://github.com/QuEST2025/speedup/blob/main/tables.pdf
>
> In addition, we conducted further experiments on the scalability of our CUDA kernels in isolation, by increasing the arithmetic intensity of the GEMM problems. The following table shows the speedup achieved with a fixed N = 4096 while varying the M (batch times sequence length) and K (hidden) dimensions.
>
> | m \ k  | 2048  | 4096  | 8192  | 12288 | 16384 |
> |--------|-------|-------|-------|-------|-------|
> | 1024   | 1.84x | 2.87x | 4.31x | 5.00x | 5.55x |
> | 2048   | 1.94x | 3.23x | 4.83x | 5.58x | 6.07x |
> | 4096   | 2.17x | 3.78x | 5.44x | 5.86x | 6.27x |
> | 8192   | 2.24x | 4.04x | 5.86x | 6.22x | 6.28x |
>
> We hope this clarifies your concerns, and are happy to continue the discussion if the reviewer finds it necessary.

---

### Official Review · Reviewer_zJfZ · 2025-03-21

**Overall Recommendation:** 4

**Summary:**

This paper explores how to improve quantization aware training.  Following recent work in post-training quantization, this paper proposes a combination of techniques it calls QuEST.  QuEST involves using a Hadamard Transform in the forward pass to improve the quantization process and introduces "trust estimation" to improve gradient estimation in the backward pass.  The evaluation shows QuEST improves the tradeoff between model performance at a given precision.  A new precision scaling law is fit to these results.  The evaluation also shows that at 4-bits the resulting model is faster in evaluating layers than when using BF16 on RTX 4090 hardware.

**Claims And Evidence:**

The claims of improved QAT results seem to rely mainly on a comparison to results from a five year old paper, LSQ, (Esser et al., 2019) as shown in Figure 3.   This does not seem to me to be sufficient evaluation to establish QuEST as a new SOTA for QAT.

The claim of improved efficiency seems somewhat orthogonal to the proposed approach (even if supported by empirical evaluation).  Using lower precision can be more efficient was, I thought, well known.  That said, it was unclear to me how well optimized the baseline in Figure 6 is (i.e., is it using cuDNN or the software flow described in Section 5, but on FP16?)

**Essential References Not Discussed:**

I didn't see AdaBin https://arxiv.org/abs/2208.08084 (ECCV 2022) cited and I thought that work was also doing quantization aware training.

I also was thinking some of the earlier works on binary neural networks ought to have been cited and discussed.  E.g.,

Matthieu Courbariaux, Yoshua Bengio, and Jean-Pierre David. Training deep neural networks with low precision multiplications. arXiv, 2014.

Matthieu Courbariaux, Yoshua Bengio, and Jean-Pierre David. Binaryconnect: Training deep neural networks with binary weights during propagations. In NIPS, 2015.

Matthieu Courbariaux, Itay Hubara, Daniel Soudry, Ran El-Yaniv, and Yoshua Bengio. Binarized neural networks: Training deep neural networks with weights and activations constrained to+ 1 or-1. arXiv, 2016.

**Experimental Designs Or Analyses:**

One objection to the paper is lack of quantitative comparison with more recent QAT training proposals.

**Methods And Evaluation Criteria:**

The benchmarks and backbone architecture employed make sense.

**Other Comments Or Suggestions:**

Some of the writing could be improved.  For example:

- In the abstract, what is meant by efficient?  how efficient?

- Page 1: It seems incorrect to refer to "Pareto-optimal frontier" as a metric.  What does it mean for 8-bit precision to be pareto-optimal for QAT?   The word Pareto does not seem to appear in Kumar et al. (2024) and I'm not sure what the authors of the present articles mean.

- The notation in Equation 1 seems non-standard.  Usually the argument to the rond to nearest operator is not between [-1,1].  Also, if there is some notion of quantization precision that is often typically made explicit in terms of number of quantization levels or number of bits used, but the formulation in Equation 1 seems to show neither.

- Perhaps I'm missing something but to me it seems the $\gamma$ in the sentence on lines 140-142 is irrelevant to the relationship in that sentence, which in any case would appear trivially true regardless of the smoothness of the loss function based only on the definition of $S_{small}$ on line 130.


*** POST REBUTTAL COMMENTS ***

Thank you for the rebuttal response.  Unfortunately, I was unable to get to this before the too short timeline set by the conference organizers and it seems like I can no longer post a response below.  So, I'll just update the text of my review with my thoughts after your response.

Thanks for clarifying LSQ as SOTA and the particular taxonomic differences (reading through the Kumar  et al. 2024 reference was also helpful in this regard).

I am still wondering where the 'trust factor' is implemented in your code.   If you are still able to respond (which I guess unfortunately you cannot), please point us to the files/lines.

Regarding the "inconsistent SGD iterations" references cited, as I understood them they show a specific approximation (e.g., sparsity with certain properties) converges, but the relationship of what the process converges to with the original (local) optimum wasn't clear to me.   For example, the description of the example in Figure 2 in Lin et al. (https://arxiv.org/pdf/2006.07253) seems to make the case that gradients for the dense network point in a wrong direction when viewed from the sparse network.  So, 'trusting' those gradients might be misleading.

Kumar et al. (2024) in Section 4.3.2 middle plot of Figure 6, "Empirical", shows 6-bits doing better than 8-bit for "Predicted" on the leftmost plot for floating point.  I see the leftmost is for integer but the caption says "predicted" (I assuming from the scaling law fit) making it unclear to me whether the 8-bit optimum point would hold up in practice (i.e., "empirically").

However, empirically, the results in your submission do seem to show an improvement to SOTA and going through the submission another time, I think I now get the intuition for why QuEST works, so I'm leaning towards raising my score.

**Other Strengths And Weaknesses:**

Section 5 provides fairly high level details of the GPU implementation.  Adding more documentation to the .zip or an appendix would make it easier to check how the provided code implements the high level algorithms presented in the paper.

Additional ablation studies would help the paper.  There is some data in Figure 5(c) showing the impact of the HT on QuEST, but this type of evaluation should be expanded.  More insight into how the 'trust factor' impacts the results could be provided through ablation studies.

**Questions For Authors:**

I'm not convinced that the 'trust factor' approach to masking out gradients is well motivated.  The data in Figure 2 shows that this masking obviously makes the resulting direction closer to a full precision gradient, but this seems to counter the goal of QAT in that the idea is to perform gradient decent accounting for the noise introduced by quantization (so the optimizations paths should differ).  If you can clarify your motivation for the trust factor, and better back that up quantitatively or theoretically, that may help the paper.  Please comment.

**Relation To Broader Scientific Literature:**

The main questions in QAT are how to quantize and how to estimate and apply the gradient.  The paper is attempting to innovate on both fronts.    The use of Hadamard Transform to improve quantization proposed in the paper seems very similar to that in recent closely related works on post-training quantization (e.g., QuIP#).  The "trust estimation" approach may be somewhat more novel.

**Theoretical Claims:**

The paper does not make specific theoretical claims.

---

> ### Author Rebuttal · Authors · 2025-04-01
>
> Thank you for the detailed review! We address all your questions below.
>
> > I'm not convinced that the 'trust factor' approach to masking out gradients is well motivated.
>
> Thank you for raising this. First, trust factor masking is motivated theoretically by directly targeting the source of error in the QAT iteration, relative to standard SGD.
>
> Specifically, the “ideal” SGD iteration is $ x_{t+1} = x_t - \nabla L( x_t) $. Instead, in QAT we execute $ x_{t+1} = x_t - \nabla L( Q(x_t) ), $ where $Q(x_t)$ is quantization. The “error” between these iterations is precisely the $ \| \nabla L( Q(x_t) ) - \nabla L( x_t ) \|$ term we seek to minimize.
>
> The theory of inconsistent SGD iterations shows that this “error” directly impacts SGD convergence: see the work of Nadiradze et al. (https://arxiv.org/pdf/2001.05918), who bounded this error by applying smoothness, and Lin et al. (https://arxiv.org/pdf/2006.07253) who investigate the same for sparse projections. In this context, our work investigates a fast heuristic for minimizing this error in the context of quantization.
>
> Second, our practical results confirm that trust factors are key to good practical convergence.
> Besides Figure 2, we also illustrated this in Appendix Figure 10, which showed that properly tuned trust factors lead to much better loss than both clipping (s = 1) and STE (large s).
>
> > There does not seem to be sufficient evaluation to establish QuEST as a new SOTA for QAT.
>
> Note that, generally, QAT methods usually fall into two categories:
> 1. General schemes that can be applied to any bitwidth (such as STE, PACT, LSQ, and QuEST).
> 2. Schemes specialized to some bit-widths, e.g. binarization, such as AdaBin.
>
> For general schemes, LSQ was SOTA: e.g., recent work from IBM & MIT (https://arxiv.org/abs/2404.03605) on outlier suppression still uses a variant of LSQ, whereas the recent work of Kumar et al. only uses STE.
> Since our work is focused on general scaling laws, we did not consider specific binarization schemes, as they do not directly “port” across bit-widths.
>
> To fully address your concern, we have ported AdaBin to our setting and compared it with QuEST at 30-100M scale.
> Remarkably, we found that QuEST consistently outperforms AdaBin in the W1A1 setting (for which AdaBin is specifically designed). Moreover, the stripped-down QuEST *without the Hadamard* recovers the performance of AdaBin:
>
> https://github.com/QuEST2025/speedup/blob/main/AdaBin.md
>
> We hope this clarifies our positioning: we believe QuEST is indeed the new SOTA for general QAT.
>
> > The claim of improved efficiency seems somewhat orthogonal.
> The goal of our kernels is showing that QuEST models can execute fast; this is not obvious since they require a Hadamard multiplication and dynamic clipping of the activations on the forward pass.
>
> > It was unclear to me how well optimized the baseline in Figure 6 is.
>
> The baseline in Figure 6 is near-optimal for BF16; please see more details in our reply to Reviewer `HH36`.
>
> **Editorial comments and comments about supplementary material.**
>
> Thank you for the detailed examination and useful editorial notes! We will address all these in the next revision, and add a more detailed README for the CUDA code structure.
>
> > The use of Hadamard Transform seems similar to PTQ (e.g., QuIP#).
>
> Please see “Regarding the Hadamard Transform (HT)” part of the reply to Reviewer `Xwq9`.
>
> > It seems incorrect to refer to "Pareto-optimal frontier" as a metric.
>
> Pareto-optimality is defined in the introduction (page 1, col.2, l.49-50), and follows Frantar et al., ICLR24. We say that an approach X (e.g., QuEST INT4) is Pareto-superior to Y (e.g., STE INT8) if X provides better accuracy at the same model size, or, symmetrically, smaller size at the same accuracy. Figure 1 shows that QuEST INT2 is Pareto-superior to BF16 pretraining, but inferior to QuEST INT4. Thus, QuEST brings the “optimal” precision in terms of accuracy-vs-size to INT4, since no other method dominates QuEST INT4 across sizes.
>
> Kumar et al. (2024) don’t use the same terminology, but their metrics are similar. E.g., in Section 4.3.2 of their paper, they solve for P*, the precision that yields minimal loss at some model size, which is the “Pareto-optimal” precision.
>
> We used a similar setting to Kumar et al. (2024) for the scaling law study, but improved significantly on their findings in terms of optimal precision via a new method: QuEST brings down the “optimal” training precision to 4bit, down from the 7-8bit precision found to be “optimal” for STE by Kumar et al. (2024).
>
> > ...the sentence on lines 140-142 is true regardless of the smoothness of the loss function
>
> Indeed, we could obtain a bound of $T^2 |S_{small}|$ just by summing over indices $k$ in $S_{small}$. We used smoothness since if $\gamma < 1$ we would get a better bound $\gamma^2 T^2 |S_{small}|$. We thank the reviewer for this note and will simplify the derivation to not use smoothness.

---

### Decision · Program_Chairs · 2025-05-01

**Decision:**

Accept (poster)

**Comment:**

This is a paper on quantized model training. Reviewer Xwq9 provides a set of valid limitations for the work, and while I agree with their concerns, I am inclined to agree overall with zjfZ that this is a solid improvement. The community would be interested in this work, and any idea that can improve over QAT (which is long-established as SOTA) is something many people would like to read.